# Spatiotemporal prediction of COVID-19 cases using inter- and intra-county proxies of human interactions

Behzad Vahedi [1✉], Morteza Karimzadeh [1] & Hamidreza Zoraghein [2]

Measurements of human interaction through proxies such as social connectedness or movement patterns have proved useful for predictive modeling of COVID-19, which is a challenging task, especially at high spatial resolutions. In this study, we develop a Spatiotemporal autoregressive model to predict county-level new cases of COVID-19 in the coterminous US using spatiotemporal lags of infection rates, human interactions, human mobility, and socioeconomic composition of counties as predictive features. We capture human interactions through 1) Facebook- and 2) cell phone-derived measures of connectivity and human mobility, and use them in two separate models for predicting county-level new cases of COVID-19. We evaluate the model on 14 forecast dates between 2020/10/25 and 2021/01/24 over one- to four-week prediction horizons. Comparing our predictions with a Baseline model developed by the COVID-19 Forecast Hub indicates an average 6.46% improvement in prediction Mean Absolute Errors (MAE) over the two-week prediction horizon up to 20.22% improvement in the four-week prediction horizon, pointing to the strong predictive power of our model in the longer prediction horizons.

[1] Department of Geography, University of Colorado Boulder, Boulder, USA. [2] Social and Behavioral Science Research, Population Council, New York, USA. ✉email: Behzad@colorado.edu

Human interaction in close physical proximity is the primary cause of the transmission of highly contagious diseases such as COVID-19[1]. Measuring human interaction is, therefore, an important step in understanding and predicting the spread of COVID-19[2,3]. However, tracking human interactions requires rigorous contact tracing at national and regional scales, which has not been implemented in the United States due to the economic, legal, and sociocultural concerns, as well as inadequate testing supplies, and insufficient national coordination[4].

As a result, researchers have adopted different proxies to track human-interaction levels and connectedness. One such proxy is the "Social Connectedness Index" (SCI), generated from Facebook's friendship data. SCI represents the probability that two users in a pair of regions (e.g., U.S. counties) are friends (i.e., connected) on Facebook[5]. Kuchler et al.[6] reported on the strong correlation between early hotspots of COVID-19 outbreak and their level of social connectedness. The underlying assumption in leveraging SCI as a proxy for physical human interactions is that individuals who are socially connected on Facebook have a higher probability for physical interaction, thereby, potentially contributing to the spread of communicable diseases. Facebook also generates measures of daily movements observed through its app, providing another measure for quantifying human interactions[7].

Human-mobility flow, as measured by anonymized cell-phone data, serves as another proxy for quantifying human interactions and the connectedness of places[8,9]. Widely used to study the spread of COVID-19, most studies incorporating cell-phone data have focused on the change in mobility within a spatial unit[10,11], while a few others have also incorporated the flow between different spatial units[12] to predict transmissions across units, albeit mostly in the early stages of the pandemic with limited evaluation data. The underlying assumption in this approach is that more movements between spatial units lead to higher interactions, and consequently, an elevated risk of disease spread.

It is unclear, however, which of these approaches—using social-media connectedness versus cell-phone-derived human-mobility flow—is a better indicator of physical interaction within and between different regions. Furthermore, the underlying assumption in each approach may not necessarily be valid in the case of COVID-19: considering the sporadic and regional stay-at-home orders across the United States, social connectedness may not lead to physical interaction, at least not to the same level as prepandemic. Similarly, given the recommended preventive measures such as mask-wearing and physical distancing[13], human flow from one location to another may not necessarily lead to physical interactions that could communicate the disease, especially in public places, where preventive measures are enforced more strictly.

In this paper, we compare the predictive power of Facebook-derived movement measurements and social connectedness, as an example of social-media proxy, with cell-phone-derived human mobility and connectedness, as an example of human-flow proxy, in forecasting county-level new cases of COVID-19 in the conterminous United States over multiple prediction horizons. To do so, we design a machine-learning model using spatiotemporally lagged variables of human-movement measurements and new COVID-19 cases weighted by the intercounty connectedness strength measured by each proxy. County-level prediction is more challenging than state-level prediction[14–16], yet it has served as the highest spatial resolution for national models in the United States since cases have been aggregated and reported at the county level. Long-term county-level predictions are also essential for policymaking and resource allocation.

The unique characteristics of COVID-19, including its presymptomatic and asymptomatic contagiousness, rapid spread,

along with variations in regional response policies, such as inconsistent and sporadic testing and contact tracing, make forecasting the spatial patterns of this disease challenging. Researchers have used a variety of methods, including time-series autoregressive models[17–19], machine-learning techniques[20–22], epidemiologic models such as the SIR model and its variants[23,24], and combinations of these methods[25] for forecasting COVID-19 incidence rates.

In this paper, we analyze five different machine-learning algorithms and use the best algorithm, i.e., the one generating the lowest-average prediction root mean squared error (RMSE) and mean absolute error (MAE) using our features, to develop a spatiotemporally autoregressive model for predicting incident (new) cases of COVID-19 at the county level in short-term (one-week ahead) and longer-term (two- to four-week ahead) horizons.

We compare our best model predictions against a baseline model as well as an ensemble model developed by the "COVID-19 Forecast hub" team[25]. The ensemble model (referred to as COVIDhub-Ensemble henceforth) is one of the most prominent collective efforts in forecasting COVID-19 in the United States and is used by the Centers for Disease Control and Prevention (CDC) to report predictions of new cases in US counties in one- to four-week ahead horizons[26,27]. The Baseline model, which we refer to as COVIDhub-Baseline henceforth, is a neutral, reference model with "a predictive median incidence equal to the number of reported cases in the most recent week"[28].

Our contributions include developing a spatiotemporally autoregressive machine-learning model for predicting county-level new COVID-19 cases that incorporates spatiotemporally lagged intercounty and intracounty predictive features. We show that this model improves on average the two- to four-week ahead (long-term) predictions of county-level new cases of COVID-19 in the coterminous United States compared with the COVIDhub-Baseline model, regardless of the specific proxy (Facebook-derived or SafeGraph-derived) used in creating the spatiotemporally lagged features. Our best model also beats the Ensemble model on average in three- and four-week ahead prediction horizons. As part of our evaluations, we also compare the predictive power of Facebook-derived and SafeGraph-derived features in the context of our models for predicting new COVID-19 cases.

## Results

**Algorithm selection**. Five different machine-learning algorithms were trained and tuned using each set of features named in Table 5, and tested over the last 14 weeks of our dataset (same dates as Supplementary Table 1), by holding out one week at a time for testing. Table 1 reports the average performance for each algorithm. EXtreme Gradient Boosting (XGB) performed better on unseen data compared with other tree-based ensemble algorithms and the neural networks, including Feed Forward Neural Network (FFNN) and Long Short-Term Memory (LSTM) network (Table 1). Therefore, we used XGB for developing short-term and long-term prediction models. The training and evaluation (testing) RMSE and MAE values of each model are reported in Table 1. The reported values are for the natural logarithm of [new cases per 10k population + 1], which we used as a transformed target variable in the models, given the skewed distribution of new cases (or new cases per 10k population) in counties.

**Comparing social media- and cell-phone-derived features**. To compare the relative strength of Facebook-derived movement and connectedness against SafeGraph-derived movement and

**Table 1 Performance comparison of machine learning regressors.**

| Model | Training RMSE | Training MAE | Evaluation RMSE | Evaluation MAE |
|---|---|---|---|---|
| Random Forest (RF) | 0.472 | 0.342 | 0.474 | 0.333 |
| Stochastic Gradient Boosting (SGB) | **0.423** | 0.316 | 0.487 | 0.329 |
| eXtreme Gradient Boosting (XGB) | 0.439 | **0.311** | **0.461** | **0.319** |
| Feed Forward Neural Network (FFNN) | 0.458 | 0.332 | 0.493 | 0.366 |
| Long Short-Term Memory (LSTM) | 1.296 | 1.024 | 1.349 | 1.304 |

The lowest value in each column (corresponding to the best performing model in each category) is bold-faced.

connectedness, as proxies for physical human interactions, we designed a set of intracounty and intercounty interaction features using each proxy and incorporated each set of features separately to develop spatiotemporally lagged autoregressive prediction models of new cases of COVID-19 (i.e., target variable). We then compared the predictions of these models against each other, as well as a temporally lagged base model that we developed (not to be confused with the COVIDhub-Baseline model that we use for the final, comparative evaluation), all of which were trained using the XGB algorithm.

Our base model incorporates a series of socioeconomic, demographic, and temperature variables, as well as temporal lags of the target variable in the same county only, thus, we call it Temporal XGB (TGXB). The SpatioTemporal XGB (STXGB) models, in addition to temporal lags, also incorporate intracounty movement features and spatiotemporal lags of the target variable weighted by the intercounty connectedness strength (connectedness is beyond geographic proximity, as defined in the "Methods" section). Specifically, the spatial lags in STXGB are the weighted average of the log-transformed target variable (Ln(weekly new cases per 10 k population + 1)) by intercounty connectedness index of connected counties, where connectedness index is calculated as either (a) Facebook Social Media Connectedness Index (SCI, in the STXGB-FB model), or (b) Flow Connectedness Index (FCI) derived from SafeGraph's cell-phone movement data, forming STXGB-SG and STXGB-SGR models (described in detail in the Methods section).

Table 2 presents the average error values of predicted total new cases and new cases per 10 k population in the one-week prediction horizon over the 14 forecast dates using the TXGB and STXGB models and Fig. 1 presents MAE and RMSE of the models when predicting total new cases over each forecast date. All variants of STXGB (-FB, -SG, and -SGR) achieved lower errors compared with TXGB, meaning that, the incorporation of movement features and spatiotemporal lags weighted by connectedness indices derived from either Facebook or Safegraph cell-phone data was advantageous across the board, compared with the temporal lags only (TXGB). Furthermore, STXGB-FB, which uses Facebook-derived features, outperformed all other models in average RMSEs and MAEs.

**Long-term predictions and evaluation against the COVIDhub-Baseline.** We compared the predictions of both STXGB-FB and STXGB-SG models (as the two best performing models, one using Facebook- and the other SafeGraph-derived features) against the predictions of the COVIDhub-Baseline model over one-, two-, three-, and four-week horizons. We trained and tuned STXGB models for each prediction horizon separately. We then used the reported new cases by Johns Hopkins University Center for Systems Science and Engineering (JHU CSSE)[29] as ground truth to calculate MAE and RMSE of each prediction, performed across all 4 prediction horizons and 14 forecast dates (56 predictions in total). The 14-week evaluation period covers a period characterized by increasing and decreasing trends, as well as the peak,

of the number of new cases in the United States. Figure 2 and Table 3 present MAE values of STXGB-FB and -SG models in comparison with the COVIDhub-Baseline model (RMSE values are presented in Supplementary Fig. 1). Our STXGB-FB model improves MAEs compared with the COVIDhub-Baseline model in two- to four-week prediction horizons, with average improvements of 6.46%, 13.32%, and 19.30% over two-week, three-week, and four-week ahead prediction horizons, respectively. STXGB-SG model also outperforms the COVIDhub-Baseline in two- to four-week horizons, with average improvements of 4.48%, 14.28%, and 20.22%, respectively (Table 3).

In the one-week prediction horizon, our STXGB-FB and STXGB-SG models outperform the COVIDhub-Baseline model on half of the forecasting dates (7 out of 14). In longer prediction horizons (2- to 4-week horizons), STXGB-FB and STXGB-SG outperform the COVIDhub-Baseline with a larger MAE margin, achieving a lower MAE in 32 and 34 predictions (forecast date/horizon combinations) out of the total 42 predictions, respectively (STXGB-FB outperforms the Baseline in 10, 10, and 12 forecast dates over two-, three-, and four-week prediction horizons, respectively; for STXGB-SG, the corresponding values are 10, 11, and 13). To further contextualize these comparisons, it is worth noting that in a study on the performance of the models submitted to the COVID-forecast hub in predicting incidence deaths, roughly half the submitted models had errors larger than the COVIDhub-Baseline model[28].

It is important to note that the COVIDhub-Baseline outperformed our models in 7 out of 8 predictions performed across all prediction horizons on December 20th, 2020, and January 3rd, 2021 forecasting dates (Fig. 2 and Table 3). The week of December 20th–December 26th marks an apparent local minimum in the daily number of new cases in the United States, with a relatively sharp decrease which could be attributed to the Christmas holidays when most counties report abnormally lower numbers (Fig. 2i). The week of January 3rd–January 9th is when the United States experienced the highest number of weekly new cases during the entire pandemic, most likely with numbers that were inflated due to latent reporting of cases observed during the New Year's holidays. Hence, both periods represent abnormal changes in the number of new cases, which might reflect a change in the behavior of reporting agencies (due to holidays or the weeks following them) as opposed to a sharp weekly increase or decrease in the number of cases.

**Prediction intervals.** To assess the uncertainty of our models, we generated 95% prediction intervals (PI) for the total number of new cases (described in the Methods section). Figure 3 shows the PIs generated by the STXGB-FB and STXGB-SG models in comparison with the county-level COVIDhub-Baseline model for one- to four-week prediction horizons. The COVIDhub-Baseline model generates narrower PIs compared with both of our models. However, in a few cases such as the three- and four-week ahead predictions on the 2020/11/1 forecasting date, the 95% PI of the COVIDhub-Baseline model does not include the observed value

**Table 2 Average RMSE and MAE of county-level predicted new cases and new cases per 10 k population over 14 forecast dates for the one-week horizon.**

| | Model | RMSE new case prediction | MAE new case prediction | RMSE new case/10 k prediction | MAE new case/10 k prediction |
|---|---|---|---|---|---|
| includes temporal lags | Base model (TXGB) | 338.32 | 69.60 | 16.88 | 10.04 |
| include spatiotemporal lags | STXGB with Facebook-derived features (STXGB-FB) | **308.49** | **64.33** | **16.23** | **9.48** |
| | STXGB with SafeGraph-derived features (STXGB-SG) | 318.00 | 64.86 | 16.41 | 9.64 |
| | STXGB with SafeGraph-derived features-rich (STXGB-SGR) | 345.04 | 66.32 | 16.37 | 9.60 |

The lowest values of each error metric (achieved by the best performing model) are boldfaced.

of the number of new cases, whereas the PIs of both of our models successfully include the observed values across all prediction horizons and forecasting dates. This indicates the strong predictive structure of STXGB models when leveraging either Facebook- or SafeGraph-derived features.

Figure 5(a–d) shows the percentage errors of the COVIDhub-Baseline model in comparison with the STXGB-FB and STXGB-SG models when predicting the total number of new cases in the coterminous United States across each prediction horizon. At each forecast date, percentage error is calculated by dividing the difference between the predicted value of total new cases and the observed value of total new cases by the observed value (positive values indicate overprediction, and negative values indicate underprediction). As seen in this figure, all models underestimate the total number of new cases before the 2020/11/15 forecasting date and overestimate this number after the 2021/01/10 forecasting date. These are respectively the periods of steady increase and steady decrease in the (7-day-averaged) trend of the total number of new cases in the United States (Fig. 2i). In the period between 2020/11/15 and 2021/01/10, when the trend of the total number of weekly new cases fluctuates, our models on average have a lower percentage error compared with the COVIDhub-Baseline in two- to four-week prediction horizons and the COVIDhub-Baseline has a lower percentage error in the one-week horizon.

**Spatial distribution of errors**. Both our models performed better than the COVIDhub-Baseline model on 7 out of 14 forecasting dates over the one-week horizon and outperformed the COVIDhub-Baseline on at least 10 out of 14 forecasting dates across all longer than one-week horizons. To find potential explanations for this inconsistency between the short-term and long-term performance of our models, we inspected the spatial patterns of errors. Figure 4 illustrates the maps of confirmed new cases per 10k population along with prediction errors per 10k population generated by the STXGB-FB model for two forecasting dates of Nov. 1 and Nov. 8, 2020. The purple-shaded counties in the error maps are those with model underestimation of new cases, and the brown shades indicate overestimations of observed values. As can be seen in this figure, the majority of counties with high prediction errors (per 10k) are located in the rural Midwest with relatively high numbers of cases per 10k population during the November surge, albeit these are counties with fewer total cases compared with more populated, urban ones. It is worth noting that we use normalized (by 10k population) maps in Fig. 4, since choropleth maps would be biased by patterns of population distribution otherwise.

Figure 4 also demonstrates clusters of apparent underestimations in Georgia and Texas on the Nov. 1 forecasting date, followed by apparent overestimations in the same areas for the week after. The opposite pattern is the case for Kentucky. In the case of Georgia, the high-error clusters can almost perfectly delineate the boundary of the state. This discrepancy could be a result of lags or different policies in testing and reporting COVID-19 cases. These potential short-term lags in reporting by some states may explain why our model performs considerably better in the longer-term prediction horizons. This indicates that our models might be in general sensitive to short-term inconsistencies in reporting, but more stable over longer-term horizons.

Additionally, in this study, we chose to evaluate our model predictions against the weekly aggregation of the raw number of cases (as opposed to smoothed values) to ensure that we are not characterizing our results with an additional advantage. The

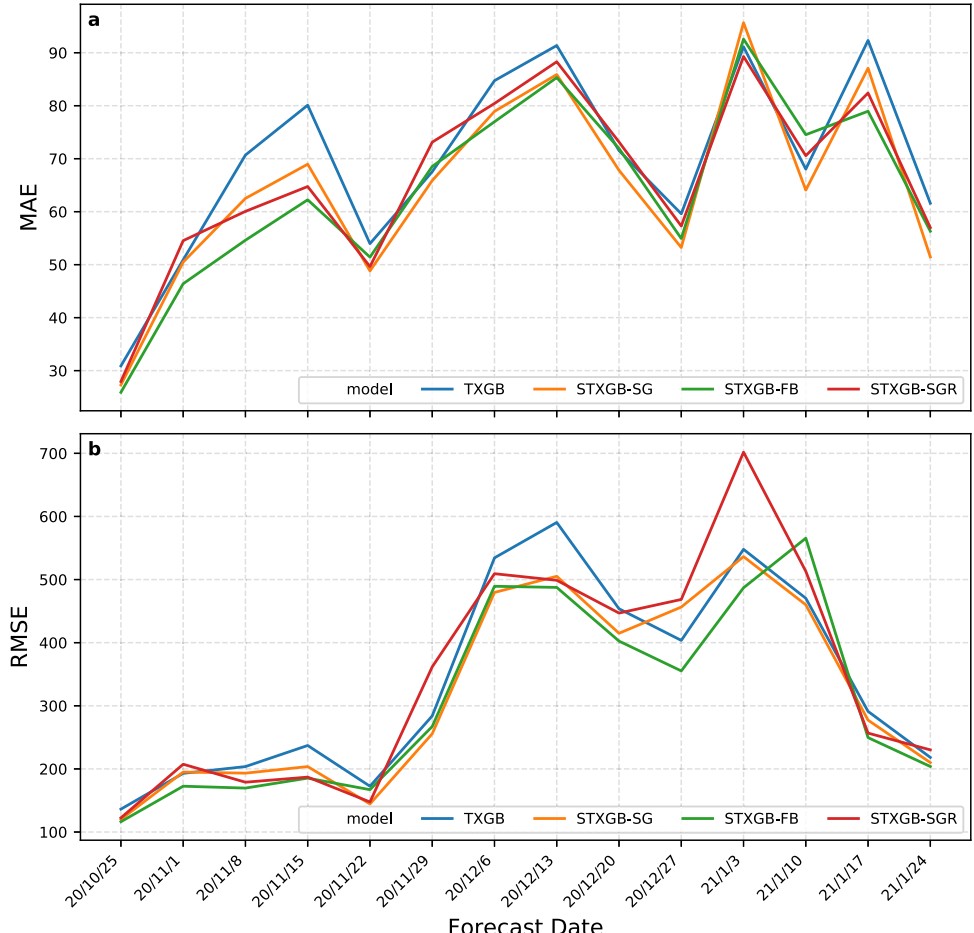

**Fig. 1 Prediction errors of the SpatioTemporal eXtreme Gradient Boosting (STXGB) models in one-week prediction horizon against a temporal autoregressive model without spatial lags (TXGB). a** Prediction MAE and **b** prediction RMSE of one-week ahead predictions of total new cases. STXGB-FB, which incorporates Facebook-derived features, including spatial lags based on Social Connectedness Index as well as Facebook-measured movement data, outperforms other models on average as indicated in Table 2.

number of cases is, however, reported inconsistently, and may, for example, contain underreporting during one week with over-reporting during the following weeks. Evaluating the models against the smoothed case numbers (e.g., using a 7-day moving average before weekly aggregation) might be a better reflection of the real performance of the models, which would give our results a performance boost.

The majority of counties in the United States are rural, which are also the ones with fewer medical resources, and where social media data or cell-phone mobility data, which underlie our models, might be less representative[30–32]. To investigate our models' performance in rural-majority counties compared with the COVIDhub-Baseline, we categorized the counties into urban- and rural-majority by calculating an urbanization index for each county (Supplementary Note 2). In total, 2391 counties (~77%) were identified as rural and 712 as urban.

We then calculated the prediction errors of the number of cases and the number of cases per 10k population for the COVIDhub-Baseline and STXGB-FB models in each category across four prediction horizons for the Nov. 8 forecasting date (Supplementary Fig. 4 presents the former case and Fig. 5e–h presents the latter). Both models generate considerably lower median errors and narrower interquartile error ranges in rural counties when predicting the total number of new cases (not normalized by population), which could be attributed to the overall higher prevalence and higher variance of COVID-19

cases in urban counties in our prediction horizons. However, the opposite is the case when predicting the number of weekly new cases *per 10k population*; both models have wider interquartile ranges in rural counties across all prediction horizons. This could be due to the overall higher prevalence of COVID-19 *per population* in rural counties during the selected prediction horizons.

As evident in Fig. 5e–h, STXGB-FB has lower prediction errors in predicting the number of new cases with a narrower interquartile range (IQR) in both urban and rural counties compared with the COVIDhub-Baseline model, across all prediction horizons, except for the shortest one (one-week horizon), which may be attributed to temporal fluctuations and policy variations in testing and case reporting as discussed above. In predicting the total number of new cases per 10k population, STGXB-FB has lower median errors in both urban and rural counties across two- to four-week prediction horizons, but the COVIDhub-Baseline model has a narrower IQR. Furthermore, the difference between the median prediction errors of both STXGB-FB and COVIDhub-Baseline in urban and rural counties, when predicting the number of cases per 10k population, is smaller compared with predicting the number of new cases (not normalized by population) (Supplementary Fig. 4). This points to the consistent performance of STXGB-FB in rural-majority counties, even though Facebook might not be as representative in these areas[30].

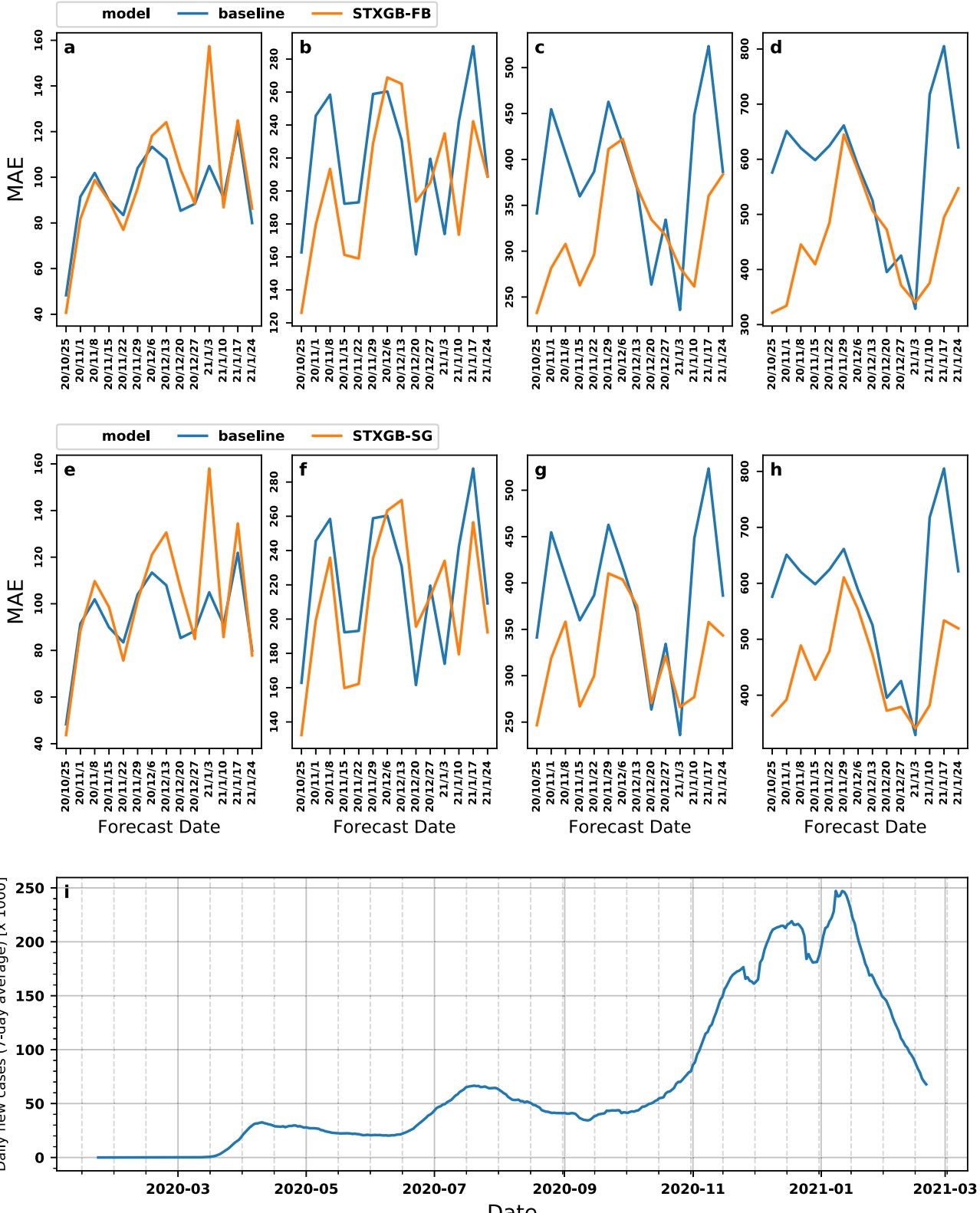

**Fig. 2 Prediction-error comparison with COVIDhub-Baseline.** Prediction MAEs of (**a**–**d**) STXGB-FB and (**e**–**h**) STXGB-SG models compared with the COVIDhub-Baseline model over four prediction horizons on 14 forecasting dates. **a** and **e** One-week horizon, and **b** and **f** two-week horizon, **c** and **g** three-week horizon, **d** and **h** four-week horizon. In each plot, MAE values are calculated as the error in predicting the total number of new cases. **i** Daily new cases in the coterminous United States. The values represent the 7-day average of total new cases reported in each day during our study period.

**Table 3 Comparison of the mean absolute prediction errors (MAE) generated by the COVID-19 Forecast Hub Baseline model and our STXGB-FB and STXGB-SG models in 1- to 4-week prediction horizons across all forecasting dates.**

| Forecast date | Model | Percentage improvement in MAE compared to COVIDhub-Baseline | | | |
|---|---|---|---|---|---|
| | | 1-week horizon | 2-week horizon | 3-week horizon | 4-week horizon |
| 2020/10/25 | STXGB-FB | 15.83 | 22.50 | 31.75 | 44.16 |
| | STXGB-SG | 9.58 | 18.73 | 27.73 | 36.88 |
| 2020/11/01 | STXGB-FB | 10.66 | 26.93 | 38.08 | 48.66 |
| | STXGB-SG | 3.20 | 18.92 | 29.86 | 39.80 |
| 2020/11/08 | STXGB-FB | 3.01 | 17.43 | 24.34 | 28.18 |
| | STXGB-SG | −7.62 | 8.76 | 11.96 | 21.18 |
| 2020/11/15 | STXGB-FB | 0.16 | 16.13 | 26.98 | 31.57 |
| | STXGB-SG | −9.66 | 16.91 | 25.81 | 28.54 |
| 2020/11/22 | STXGB-FB | 7.78 | 17.64 | 23.40 | 22.35 |
| | STXGB-SG | 9.31 | 16.03 | 22.49 | 23.30 |
| 2020/11/29 | STXGB-FB | 8.52 | 11.61 | 11.15 | 2.51 |
| | STXGB-SG | 2.49 | 9.01 | 11.34 | 7.66 |
| 2020/12/06 | STXGB-FB | −4.18 | −3.27 | −1.19 | 1.46 |
| | STXGB-SG | −6.74 | −1.14 | 3.26 | 5.87 |
| 2020/12/13 | STXGB-FB | −14.94 | −14.79 | −0.36 | 3.54 |
| | STXGB-SG | −20.93 | −16.74 | −1.88 | 9.91 |
| 2020/12/20 | STXGB-FB | −21.13 | −19.84 | −26.90 | −19.51 |
| | STXGB-SG | −24.94 | −21.10 | −2.66 | 5.86 |
| 2020/12/27 | STXGB-FB | −0.08 | 6.66 | 5.10 | 12.68 |
| | STXGB-SG | 3.89 | 2.97 | 3.92 | 10.88 |
| 2021/01/03 | STXGB-FB | −50.03 | −35.10 | −19.49 | −3.58 |
| | STXGB-SG | −50.57 | −34.55 | −12.83 | −3.68 |
| 2021/01/10 | STXGB-FB | 4.77 | 28.44 | 41.67 | 47.67 |
| | STXGB-SG | 5.98 | 25.98 | 38.21 | 46.77 |
| 2021/01/17 | STXGB-FB | −2.47 | 15.83 | 31.16 | 38.58 |
| | STXGB-SG | −10.30 | 10.88 | 31.64 | 33.72 |
| 2021/01/24 | STXGB-FB | −7.93 | 0.26 | 0.75 | 11.94 |
| | STXGB-SG | 2.65 | 8.06 | 11.13 | 16.44 |
| **Average Pct.** | **STXGB-FB** | **−3.57** | **6.46** | **13.32** | **19.30** |
| **Change** | **STXGB-SG** | **−6.69** | **4.48** | **14.28** | **20.22** |

The average values for each model and across each horizon are bold-faced.

**Comparison with the forecast Hub's ensemble model**. In addition to the COVIDhub-Baseline model, we compared our models against the COVIDhub-Ensemble model, which is an ensemble of 32 models that regularly submit forecasts to the COVID-19 Forecast hub[25], and is used by the Centers for Disease Control and Prevention (CDC) to report predictions of new cases in U.S. counties in one- to four-week ahead horizons[26,27].

In comparison with the Ensemble model, STXGB-SG achieves lower average prediction MAEs (over the 14 forecast dates) in the three- and four-week prediction horizons, and a higher average MAE in the one- and two-week horizons. The STXGB-FB model also outperforms the Ensemble model on average in the four-week prediction horizon (Table 4 and Supplementary Table 3). This is noteworthy because an analysis of 23 models that submitted forecasts of weekly COVID-19 mortality counts to the COVID-19 Forecast Hub, published by the Hub[28], reported that "forecasts from all models showed lower accuracy and higher variance as the forecast horizon moved from 1 to 4 weeks ahead". Furthermore, the same analysis reported that the Ensemble model was "consistently the most accurate model when performance was aggregated by forecast target". However, as mentioned, our STXGB-SG beats the Ensemble when forecasts are aggregated over three- and four-week ahead horizons (during the 14 evaluation weeks).

STXGB-SG outperforms the Ensemble in terms of prediction MAE in 8, 10, and 8 forecast dates (out of 14 forecast dates) over two-, three-, and four-week horizons respectively. The corresponding numbers for STXGB-FB are 8, 9, and 8 forecast dates. Due to space limitations, here we only present the average MAE comparison. Detailed performance comparisons between STXGB models and the COVIDhub-Ensemble model are presented in Supplementary Note 4. Supplementary Fig. 2 and 3 present the prediction intervals of STXGB models and the COVIDhub-Ensemble model and Supplementary Table 3 presents the prediction MAE values of each model across the four prediction horizons and for each forecast date.

## Discussion

We demonstrated that incorporating (1) spatiotemporal lags using intercounty indices of connectedness and (2) intracounty measurements of movement improves the performance of high-resolution COVID-19 predictive models, especially over long-term horizons. Short-term and long-term predictions of COVID-19 cases help the federal and local governments make informed decisions such as imposing or relaxing business restrictions or planning resource allocation in response to the forecasted trends of COVID-19[33,34].

By incorporating the aforementioned spatiotemporal lags, the STXGB-FB and STXGB-SG models outperformed the COVIDhub-Baseline model in two-, three-, and four-week prediction horizons on average, with inconsistent comparisons in the one-week horizon. Our error maps suggest that this inconsistency might be partly due to inconsistent and delayed testing and reporting by some states. Furthermore, the STXGB-SG model achieved a lower-prediction MAE on average compared with the Ensemble model currently used by the CDC in reporting county-level new cases of COVID-19 in three- and four-week prediction horizons.

It is important to note that our conclusions are valid at the county level and not for any specific region, age group, or gender. Furthermore, they only apply to our study design, time, and period of evaluation. We hope that this work invites other researchers to investigate the power of similar spatiotemporal lags in predictive models in different parts of the world.

The superiority of STXGB over purely temporal models (such as our base TXGB) points to the importance of incorporating both within-unit (e.g., intra-county) and between-unit (e.g., intercounty) interactions when predicting a highly contagious disease such as COVID-19. Predictive models that focus on within-state boundaries are likely to underperform because after all, the disease does spread across geographic unit borderlines.

Our results showed that using spatiotemporal lags of either Facebook-derived or SafeGraph-derived features, implemented in STXGB-FB and STXGB-SG models, respectively, and within the same architecture, generates lower prediction errors on average compared with the COVIDhub-Baseline model in prediction horizons longer than one week (Table 3).

Furthermore, the STXGB-FB model performed better than the STXGB-SG model on average in one- and two-week prediction horizons, whereas the latter outperformed the former in three- and four-week prediction horizons (on average). This points to the predictive power of our model structure as well as spatio-temporally lagged connectedness and movement feature, independent of the specific datasets that we used as a proxy for measuring human interaction and movement.

It is worth noting that both models use movement as features (in addition to connectedness) derived from each dataset as explained in the Methods section. However, to maintain compatibility, we used a formulation similar to Facebook's Social

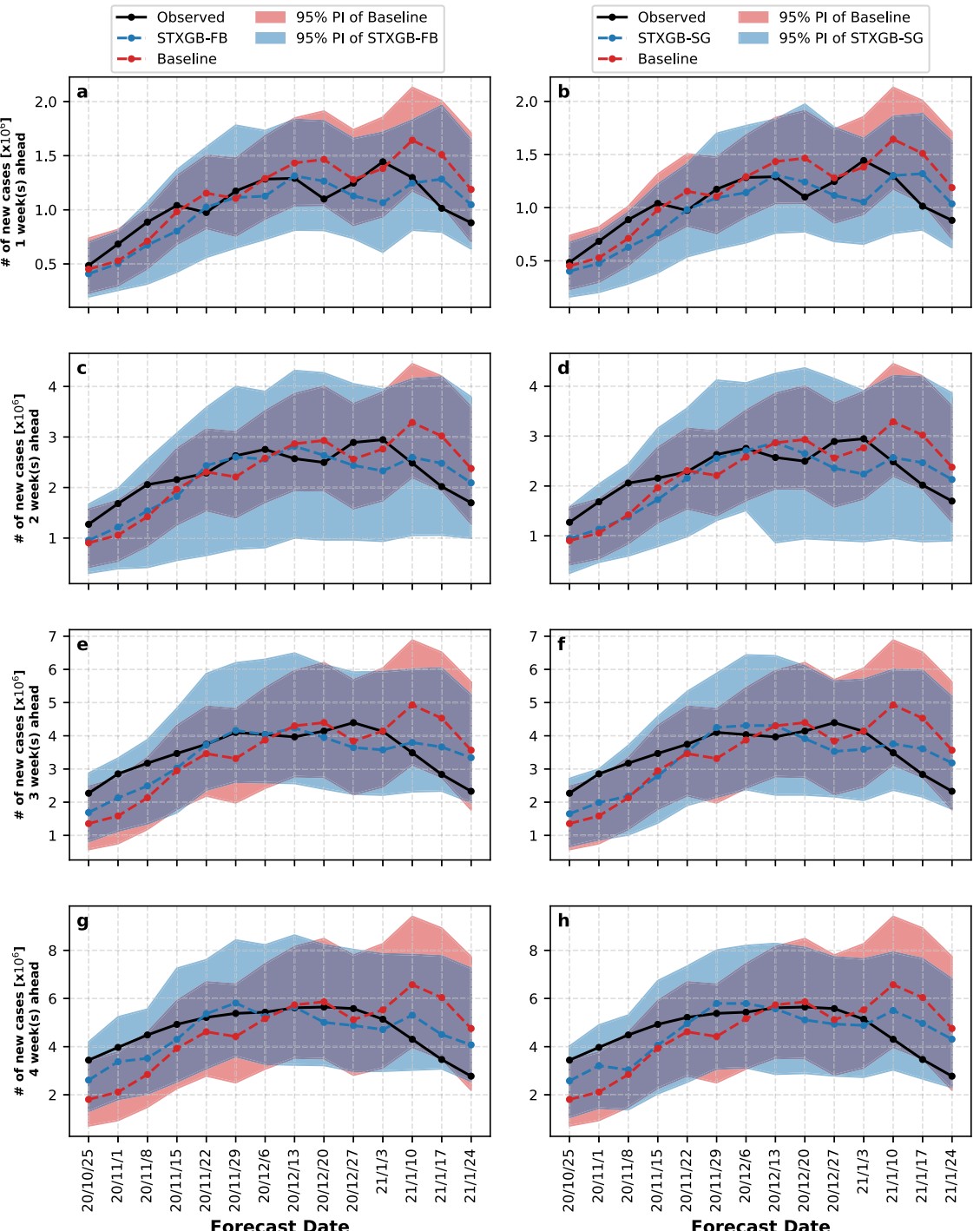

**Fig. 3 About 95% prediction interval of the STXGB models compared with the COVIDhub-Baseline over four prediction horizons.** The panels on the left show the predictions of the total number of cases and 95% PIs of STXGB-FB (blue dashed lines and regions, respectively) compared with the COVIDhub-Baseline (red dashed lines and regions). The panels on the right show the predictions of the total number of cases and 95% PIs of STXGB-SG compared with the COVIDhub-Baseline using a similar color scheme. The solid black lines represent the total number of observed cases at each forecast date. **a** and **b** One-week horizon, **c** and **d** two-week horizon, **e** and **f** three-week horizon, and **g** and **h** four-week horizon.

Connectedness Index when creating a corresponding index from SafeGraph data (which we call Flow Connectedness Index, refer to the Methods section). This might have had adverse effects on the predictive power of the cell-phone-derived features. Nevertheless, the resulting model performs better than the COVIDhub-Baseline in long-term predictions on average (Table 3). We will investigate alternative designs of intercounty connectedness metrics from SafeGraph mobility data in the future to ensure the

utilization of the full potential of this dataset. However, Safe-Graph discontinued the publication of its Social Distancing Dataset as of April 15, 2021. Our alternative approach of using social media data provides an additional pathway for predictive modeling of COVID—as evidenced by our quantitative evaluations.

Supplementary Note 5 presents detailed information on the interpretability of our models. Specifically, we analyzed the

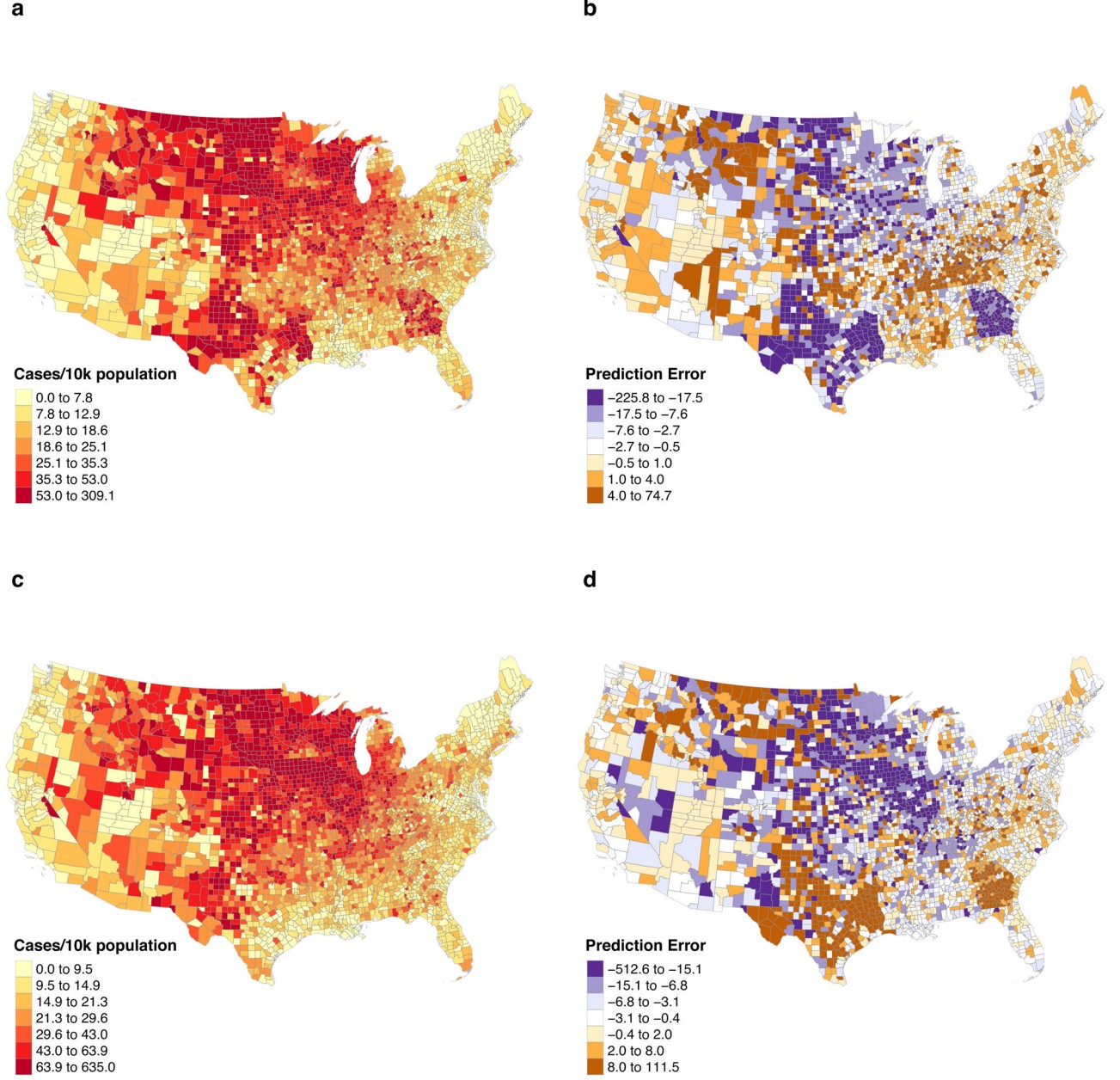

**Fig. 4 A map of COVID-19 cases per 10k population and errors in predicting them. a** The number of confirmed new cases per 10k population over the week ahead of forecasting date Nov. 1, 2020. **b** Prediction errors for the same forecasting date. **c** Number of new cases over the week ahead of the Nov. 8, 2020 forecasting date. **d** Prediction errors for the same forecasting date. The pattern of errors in Georgia, Texas, and Kentucky flips from Nov. 1 to Nov. 8, indicating potential lags in testing and reporting. The purple-shaded counties in the error maps are those with model underestimation of new cases, and the brown shades indicate overestimations of observed values.

importance of all the features used in the STXGB-FB and STXGB-SG models, across all forecasting dates and prediction horizons. Based on this analysis, the importance of the Social Proximity to Cases (SPC) in the STXGB-FB model is higher than its counterpart (Flow Proximity to Cases or FPC) in the STXGB-SG model. One-week lagged change in SPC constantly has the second-highest importance in STXGB-FB across all 56 forecast dates/prediction horizons (one-week lagged change in incidence rate has the highest importance). On the contrary, one-week lagged change in FPC is the second most important feature in only 11 of 56 predictions. The importance of movement-related features is relatively higher in STXGB-SG compared with STXGB-FB. Supplementary Fig. 5 and 6 demonstrate the feature

importance of STXGB-FB and STXGB-SG models, respectively, for a snapshot on Nov. 8 forecasting date.

Furthermore, Supplementary Note 6, Supplementary Table 4, and Supplementary Fig. 7 and 8 present a comparison between STXGB-FB and -SG models and the COVIDhub-Baseline model when predicting the number of cases in 50 counties with the highest numbers of weekly new COVID-19 cases.

While our models predict the number of new cases (via incidence rates), the number of hospitalizations or deaths (mortality rate) can also be considered as potential target variables. STXGB model can be modified to predict the number of deaths or hospitalizations using the same spatiotemporal feature structure. We leave the evaluation of these models for future research.

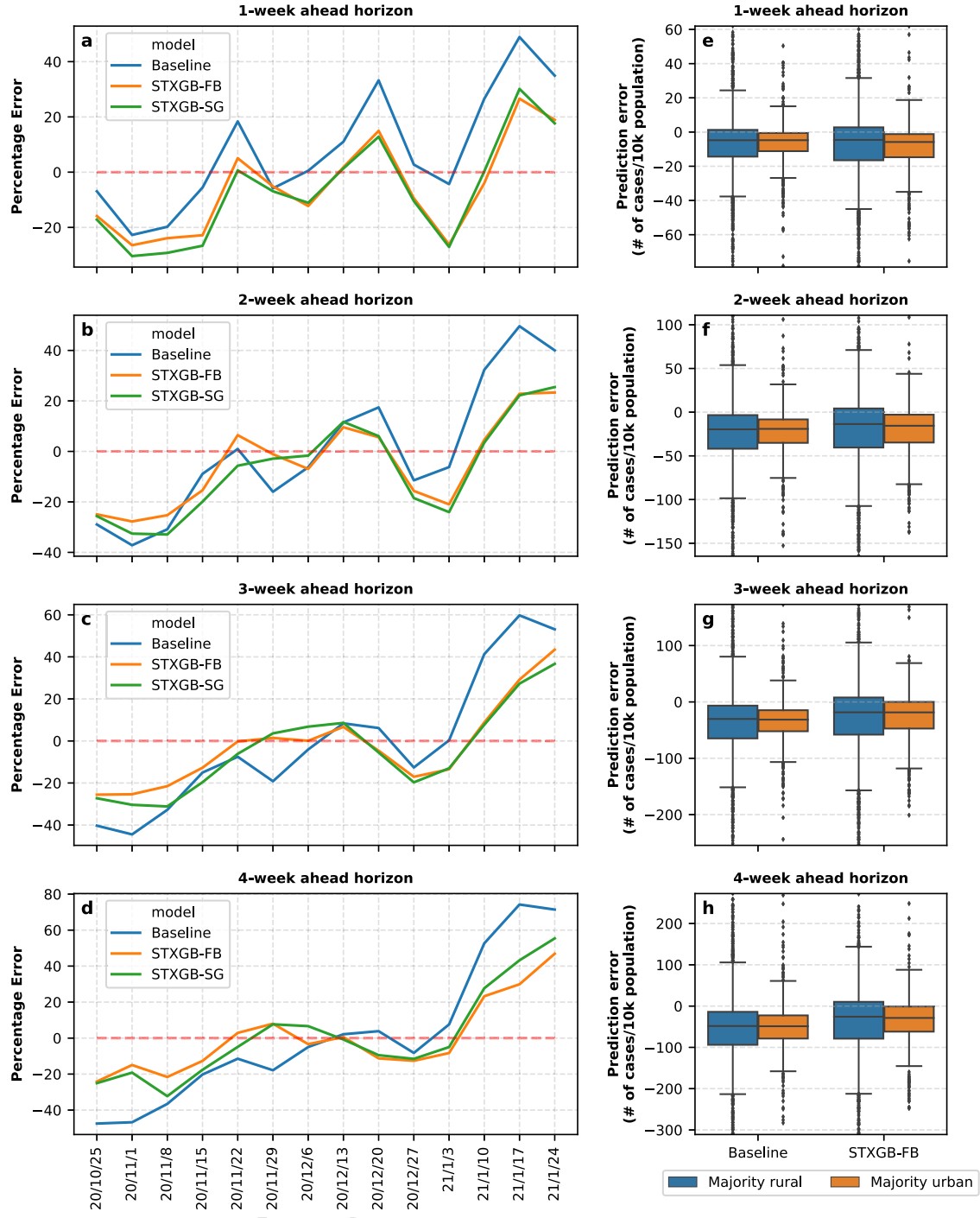

**Fig. 5 Percentage error comparison in predicting total new cases and prediction errors in urban vs. rural counties. a–d** The percentage error of STXGB-FB, STXGB-SG, and COVIDhub-Baseline models when predicting the total number of new cases in each prediction horizon. The percentage error is calculated by dividing the difference between the predicted and observed value by the observed value of total (in all counties) new cases. **a** One-week, **b** two-week, **c** three-week, and **d** four-week ahead prediction horizon. **e–h** Prediction errors of the number of new cases per 10k population in rural and urban counties on the Nov. 8 forecast date across four prediction horizons. The higher and lower 1% of counties is trimmed from the plot view. Data are presented as means ± SEM (n = 2391 for majority rural group and n = 712 for majority urban group) and whiskers represent 1.5 IQR.

The findings of this paper also suggest that Facebook's Social Connectedness Index can be used for predictive modeling of COVID-19 in data-poor countries without cell-phone-derived movement datasets, assuming that the Facebook usage in those countries is of comparable size and representativeness to the United States[35]. With more than 2.5 billion active users globally, Facebook provides social connectedness for many countries.

Conversely, human-mobility data through cell-phone companies, to the best of our knowledge, are available in a far fewer number of countries.

It is worthwhile to mention that Facebook (Daily Movement and Social Connectedness) or SafeGraph (Social Distancing Metrics) datasets may not be perfectly representative of different demographic groups such as the elderly, the less-affluent

**Table 4 Average percentage improvement in prediction MAEs achieved by our STXGB models compared to COVIDhub-Ensemble model over four prediction horizons.**

| Average of Pct. Change | 1-week horizon | 2-week horizon | 3-week horizon | 4-week horizon |
|---|---|---|---|---|
| STXGB-FB | −16.27 | −6.36 | −0.08 | 4.58 |
| STXGB-SG | −19.40 | −8.26 | 1.61 | 6.27 |

population, or the population living in rural areas[36]. While our predictions are county-level and not age- or gender-specific, we presented our analyses of the performance of our models in rural-majority counties, indicating better performance compared with the Baseline in the same counties. Additionally, Facebook has more than 190 million active users in the United States, and SafeGraph aggregates data from 45 million mobile devices, or approximately 10% of devices in the United States[37]. Also, according to a Pew Research Center survey conducted in fall 2016, 84% of American households contain at least one smartphone, and a third of Americans live in a household with three or more smartphones[38]. Therefore, even though social media and cell-phone data do not perfectly represent the US population, they still provide a valuable sample.

Furthermore, each dataset might contain a certain amount of noise as fake connections (generated from fake profiles or bots) in the case of Facebook, or multiple counts of the same person carrying multiple mobile devices in the case of SafeGraph. Machine-learning algorithms alleviate this issue to some degree by learning the regularities in the data (instead of noise), i.e., by treating such information as noise when trained on the observed patterns of disease spread, especially when the overwhelming majority of observed datapoints are not affected by noise[39].

As evident in Table 1, the XGB and SGB methods performed better than other machine-learning-based algorithms. A potential reason for the lower performance of FFNN and LSTM on our feature structure is the relatively small size of the training data (43 weeks of observation at most). Neural networks' main advantage is in their ability to learn features from data[40], however, they require higher amounts of training data compared with tree-based models for optimizing the model's parameters. Our results are a testament to the advantages of high-performance tree-based ensemble algorithms such as XGB with more limited training data, especially if features are well-engineered. Our spatiotemporal-lag features provide a template for such features to improve machine-learning-based predictive modeling of infectious diseases.

It is also worth noting that between XGB and SGB, SGB generated lower-average training errors when using the base and SafeGraph-derived features, but XGB outperformed SGB in testing RMSE and MAE across all 4 models. This is somewhat expected, as XGB uses the second-order derivatives of the loss function for optimization, and more importantly, a regularized-model formalization to control overfitting, which is otherwise a disadvantage with regression trees[41]. This regularized model results in better performance on unseen data. To ensure consistency, we ran all regression methods 10 times, and XGB had lower testing errors compared with all other regression methods in all 10 runs.

## Methods
This section outlines the details of feature engineering, algorithm selection, and implementation of spatiotemporal autoregressive machine-learning models for predicting new cases of COVID-19 in the conterminous United States. We describe our experimental setup for comparing the predictive power of Facebook-derived features and SafeGraph's cell-phone-derived mobility features (as proxies for human physical interaction) for this purpose, and the evaluation of our models against the COVIDhub-Baseline and COVIDhub-Ensemble models.

**Base features.** We engineered features for machine learning that can be categorized into five groups: (A) a set of county-level demographic and socioeconomic features, (B) minimum and maximum temperatures of inhabited areas in counties, (C) temporally lagged (i) weekly average and (ii) weekly change in cumulative incident rates (change in cumulative COVID-19 cases per 10k population) in each county, (D) Facebook-derived features of (i) intracounty movement measurements and (ii) exposure to COVID-19 through intercounty connectedness, and (E) SafeGraph-derived features of (i) intracounty movement measurements and (ii) exposure to COVID-19 through intercounty connectedness.

Socioeconomic, demographic, and climatic variables are shown to be correlated with the spread of COVID-19[22,42–45], therefore, we include category-A and -B features in all of our models to control for these factors. Supplementary Note 1 outlines the detailed methodology for generating features in these categories.

Features in category C indirectly capture population compartments of the susceptible–infected–recovered (SIR) epidemiological models[46,47]. We defined the COVID-19 incidence rate of a county as its number of cumulative cases per 10,000 population and included a four-week lagged ($t$-4) weekly average of incidence rates in each county as a feature (category C.i) to capture the Susceptible and Recovered compartments. If the feature value is small, many individuals in the unit have not yet contracted the disease, and therefore, are still susceptible. If the feature value is sufficiently large (level of sufficiency is learned by the model), the compartment is approaching higher levels of immunity as a whole. We use machine-learning algorithms that are capable of learning such nonlinear relationships.

To account for the latency associated with the effects of temperature, new and historical incidences, and human interaction on the spread of COVID-19, we generated four temporal weekly lags of features in categories B–E (we assume that the features in category A are static during our study period). Notably, for category C, we also included (natural log-transformed values of) change in cumulative incidence rate ($\ln(\Delta$ incidence rate $+ 1)$), i.e., the difference between the observed start-of-week and end-of-week cumulative incidence rates, during the four-weekly temporal lags ($t$-1,...,$t$-4), as another set of features in all of our models (category C.ii) (since cumulative rates are used, the "change" will always be greater than zero). These features conceptually capture the Infected compartment in the SIR model, i.e., the currently infected populations in the spatial unit. Our autoregressive modeling with multiple temporal lags allows the models to learn the rate of spread in a unit, as well as the varying incubation periods of the disease in relation to the change in temperature and demographic features[48,49].

Features in categories D and E also model the SIR population compartments, but in connected counties (through either social connectedness or flow connectedness). Features in category D represent the social media proxy (of physical human interactions), whereas features in category E represent the cell-phone-derived human mobility flow proxy.

The temperature variables, (spatio)temporally lagged change in incident-rate variables and daily-movement variables (using both Facebook and SafeGraph data) (Table 5), can capture the potential seasonality in the target variable to some extent. In addition, part of the variations observed in the daily-movement interaction data and/or the temporally lagged changes in incident rate could be the result of policy interventions or individual behavior change, captured in (training) data. Therefore, our models are capable of capturing changes in the season, policy, or trends, with nonlinear XGB models applied to these spatiotemporal features of movement, temperature, and lagged incidence rates.

We evaluate the predictive power of the Facebook-derived features (category D) and the SafeGraph-derived features (category E) against a base model by developing four different model setups (not to be confused with the final evaluations against the COVIDhub-Baseline model). Here, we provide an outline of these models, with more details on the specific algorithms and features mentioned in Table 5 in the following sections.

The first model (base model) only includes base features: socioeconomic features (category A), four temporally lagged weekly temperature features (category B), and four temporally lagged weekly changes in cumulative incidence rates in each county, as well as the weekly average of incidence rates during the fourth lagged week (category C). Therefore, the base model only incorporates temporal lags of the features and the target variable in predicting new cases of COVID-19.

The second model, which we identify by the "-FB" suffix (to note the inclusion of Facebook features), includes the base features as well as category-D features, i.e., Facebook-derived intracounty movement features (percentage of Stay Put and Change in Movement) and inter-county spatiotemporal lags of the target variable, i.e., exposure to COVID-19 through social connectedness (Social Proximity to Cases), across four temporal lags. The third model, which we identify by the "-SG" suffix, is conceptually similar to the -FB model, but with features derived from SafeGraph cell-phone mobility data instead of Facebook data. Specifically, the -SG models include the base features in addition to intercounty spatiotemporal lags of the target variable, i.e., exposure to COVID-19 through human-flow connectedness (which we call Flow Proximity to Cases), and a subset of category-E SafeGraph-derived intra-county movement features, across four temporal lags.

To explore the full potential of the movement features provided by the SafeGraph Social Distancing Metrics (SDM) dataset, we developed a fourth model,

**Table 5 The complete list of features, their temporal lags, and the models in which they are used. *t*-1, *t*-2, *t*-3, and *t*-4 indicate one-, two-, three-, and four-week lags, respectively. The target variable for one week-ahead prediction horizon on forecast date d is the number of new cases per 10k from the forecast date through the end of prediction horizon t in each county, i.e., $\Delta$(new incidence rate)$_{t,d}$. The target for the two-week-horizon prediction is $\Delta$(new incidence rate)$_{t+1,d}$, three-week-horizon is $\Delta$(new incidence rate)$_{t+2,d}$, and four-week horizon is $\Delta$(new incidence rate)$_{t+3,d}$. Ln in the table indicates natural logarithm, mean indicates weekly average, $\Delta$ indicates weekly change, i.e., difference (calculated by subtracting the value of the feature at the beginning of the week from its value at the end of the week) and Slope indicates the slope of a fitted linear regression model to the standardized daily measures of metric value as the dependent variable and standardized day of the week as the independent variable.**

| Category | Model(s) | Variables | Temporal Lag |
|---|---|---|---|
| A- socioeconomic and demographic | All of the models | population density; pct. of African American population; pct. of the male population; pct. of the population aged >65; pct. of Hispanic population; pct. of the rural population; pct. of Native American population; median household income; pct. of the population with a college degree; pct. of the population who voted republican (in 2016 election) | None (constant) |
| B- Temperature | All of the models | mean (daily minimum temperature)$_t$; mean (daily maximum temperature)$_t$ | *t*-1, *t*-2, *t*-3, *t*-4 |
| C- COVID-19 incidence rate | All of the models | Ln ($\Delta$ cumulative incidence rate $_t+1$) | *t*-1, *t*-2, *t*-3, *t*-4 |
| | | Ln (mean (cumulative incidence rate)$_t$ +1) | *t*-4 |
| D- Facebook | -FB model | $\Delta$ SPC$_t$ | *t*-1, *t*-2, *t*-3, *t*-4 |
| | | mean (SPC)$_t$ | *t*-4 |
| | | mean and slope (Stay Put)$_t$ | *t*-1, *t*-2, *t*-3, *t*-4 |
| | | mean and slope (Change in Movement)$_t$ | *t*-1, *t*-2, *t*-3, *t*-4 |
| E- SafeGraph | -SG and -SGR models | $\Delta$ FPC$_t$ | *t*-1, *t*-2, *t*-3, *t*-4 |
| | | mean (FPC)$_t$ | *t*-4 |
| | | mean and slope (pct. completely_home_device_count)$_t$ | *t*-1, *t*-2, *t*-3, *t*-4 |
| | | mean and slope (baselined distance_traveled_from_home)$_t$ | *t*-1, *t*-2, *t*-3, *t*-4 |
| | -SGR model | mean and slope (baseliend median_home_dwell_time)$_t$ | *t*-1, *t*-2, *t*-3, *t*-4 |
| | | mean and slope (baselined pct. full_time_work_behavior)$_t$ | *t*-1, *t*-2, *t*-3, *t*-4 |

in which two additional mobility-related measurements provided in the SDM dataset (that are least correlated with other features in category E) are added to the -SG model. This model thus includes categories A–C and all features in category E and is identified by the "-SGR" suffix.

**Features derived from Facebook**

*Intracounty movement features.* Facebook publishes the Movement Range dataset for 14 countries[50] and it includes two metrics called "Change in Movement" and "Stay Put", each providing a different perspective on movement trends as measured by mobile devices carrying the Facebook app. The Change in Movement metric for each county is a measure of relative change in aggregated movement compared with the baseline of February 2–February 29, 2020 (excluding February 17, 2020, President Day holiday in the United States)[50]. The Stay Put metric measures "the fraction of the population that have stayed within a small area during an entire day"[50]. We used four temporal lags of weekly averages and slopes of each metric as a feature in our -FB model. We calculated the slopes by fitting a linear-regression model to the metric value as the dependent variable and day of the week as the independent variable, both transformed to standard scale $N(0,1)$. The slope feature characterizes the overall trend in a week, as compared with the baseline period.

*Intercounty features and spatial lag modeling.* The intracounty features capture the intrinsic movement-related characteristics of a county and ignore its interactions (i.e., spatial lags) with the counties to which it is connected. Therefore, we calculated intercounty metrics of connectivity as a basis for incorporating spatio-temporal lags in our models. Notably, the connectedness in this context transcends spatial connectedness in the form of mere physical adjacency.

Social Connectedness Index (SCI), another dataset published by Facebook, is a measure of the intensity of connectedness between administrative units, calculated from Facebook friendship data. Social connectedness between two counties $i$ and $j$ is defined as[51]:

$$\text{Social connectedness(SC)}_{i,j} = \frac{\text{FB Connections}_{i,j}}{\text{FB Users}_i * \text{FB Users}_j} \quad (1)$$

where $\text{FB Connections}_{i,j}$ is the number of friendships between Facebook users who live in county $i$ and those who live in county $j$, while $\text{FB Users}_i$ and $\text{FB Users}_j$ are the total number of active Facebook users in counties $i$ and $j$, respectively. Social Connectedness is scaled to a range between 1 and 1,000,000,000 and rounded to the nearest integer to generate SCI, as published by Facebook[52]. Therefore, if the SCI value between a pair of counties is twice as large as another pair, it means the users in the first county pair are almost twice as likely to be friends on Facebook than the second county pair[51]. We used the latest version of the SCI dataset (at the time of our analyses), which was released in August 2020[52].

While SCI provides a measure of connectivity, our goal is to capture the spatiotemporal lags of COVID-19 cases in county $i$, i.e., the number of recent COVID-19 cases in other counties connected to county $i$. Using SCI, Kuchler et al.[6] created a new metric, called Social Proximity to Cases (SPC) for each county, which is a measure of the level of exposure to COVID-19 cases in connected counties through social connectedness. We use a slight variation of SPC, defined as follows for county $i$ at time $t$:

$$\text{Social Proximity to Cases(SPC)}_{i,t} = \sum_j \text{Cases Per } 10k_{j,t} \times \frac{\text{Social Connectedness}_{i,j}}{\sum_h \text{Social Connectedness}_{i,h}} \quad (2)$$

where $\text{Cases Per } 10k_{j,t}$ is the number of COVID-19 cases per 10k population (i.e., incidence rate) in county $j$ as of time $t$. For county $i$, the sums $j$ and $h$ are over all counties. In other words, SPC for county $i$, in time $t$, is the average of COVID-19 incidence rates in connected counties weighted by their social connectedness to county $i$, i.e., the spatial lag of incidence rates. To the best of our knowledge, SPC data have not been published, but we were able to generate this feature using the original method[6], modified for our weekly temporal lagged features, and calculated using incidence rates (cases per 10k population) rather than the total number of cases. In the -FB models (Table 5), we incorporated features of weekly change ($\Delta$) in SPC at four temporally lagged weeks (difference between the end and start of the lag week) to model the Infected SIR compartment in connected counties, as well as the weekly average of SPC in the fourth lagged week (*t*-4), to capture the Susceptible and Recovered SIR compartments in connected counties, similar to the rationale for features in category C, as explained earlier.

**Features derived from SafeGraph**

*Intracounty movement features.* To generate movement features from cell-phone data, we used SafeGraph's SDM dataset that is "generated using a panel of GPS pings from anonymous mobile devices"[53]. The SDM dataset contains multiple mobility metrics published at the Census Block Group (CBG) level. Among these metrics, distance_traveled_from_home (median distance traveled by the observed devices in meters) and completely_home_device_count (the number of devices that did not leave their home location during a day)[53] are conceptually closest to the metrics included in Facebook's Movement Range Dataset. We used these two features in our -SG model, which is the conceptual equivalent of the -FB model, but with cell-phone-derived features instead of the Facebook-derived features (Table 5).

We included the SafeGraph's median_home_dwell_time (median dwell time at home in minutes for all observed devices during the period), and full_time_work_behavior_devices (the number of devices that spent more than 6 hours at a location other than their home during the day)[53] in addition to the

previous two features in the -SGR model to take fuller advantage of the metrics available in the SDM dataset.

We derived baselined features from the SDM metrics as such: to address the potential effect of fewer cell-phone observations in some CBGs, we used a Bayesian hierarchical model[54,55] with two levels (states and counties), and then smoothed the daily measurements using a seven-day rolling average to reduce the effect of outliers in the data. We then aggregated CBG-level completely_home_device_ count and full_time_work_behavior_devices values up to the county level, divided by the total device_count in the county on the same day. For full_time_work_ behavior_devices, we subtracted the final proportion from the February 2020 baseline of the same metric. For the median_home_dwell_time and distance_traveled_from_

home variables, we calculated the weighted mean (by CBG population) of values per county and then calculated the percent of change compared with the February baseline.

We used weekly averages and slopes (calculated by fitting a linear-regression model to the values as the response variable and day of the week as the independent variable) of these four metrics as features in our models (Table 5).

*Intercounty features and spatial-lag modeling.* Building on the conceptual structure of SCI, we derived a novel and daily intercounty connectivity index from Safe-Graph's SDM dataset to quantify connectedness between counties based on the level of human flow from one county to the other (measured through cell-phone pings). We call this index "Flow Connectedness Index" (FCI). Using FCI, we then calculated a spatial lag metric that we call "Flow Proximity to Cases" (FPC) for each county. FPC captures the average of COVID-19 incidence rates in connected (by human movement) counties weighted by the FCI. Again, it is worth noting that connectedness in this sense goes beyond the physical connectivity of counties, and considers daily human interactions between them as the basis for determining connectivity. The similar formulations of FCI and SCI, as well as FPC and SPC, allow for direct comparison of the two networks (i.e., FB's friendship network and SafeGraph's human-flow network) in their capability to capture intercounty physical human interactions, and subsequently, to predict new COVID-19 cases.

The SafeGraph's SDM contains the number of visits between different CBGs. We aggregate these values to the county level to measure the daily number of devices that move (flow) between each county pair. Leveraging these flow measurements, we define Flow Connectedness Index (FCI) as

$$\text{Flow connectedness index}(\text{FCI})_{i,j} = \frac{\text{Device flow}_{i,j} + \text{Device flow}_{j,i}}{\text{Device count}_i * \text{Device count}_j} \quad (3)$$

where for counties $i$ and $j$, Device flow$_{i,j}$ is the sum of visits with origin $i$ and destination $j$. Device count$_i$ is the number of devices whose home location is in county $i$. We then scale FCI to a range between 1 and 1,000,000,000.

We defined FPC as

$$\text{Flow Proximity to Cases}(\text{FPC})_{i,t} = \sum_j \text{Cases Per } 10k_{j,t} \times \frac{\text{Flow Connectedness}_{i,j}}{\sum_h \text{Flow Connectedness}_{i,h}} \quad (4)$$

where Cases Per $10k_{j,t}$ is the number of confirmed COVID-19 cases per 10k population in county $j$ at time $t$, and Flow Connectedness$_{i,j}$ is the value of FCI between county $i$ and $j$.

Facebook's social network and friendship connections do not change significantly over time, and therefore, SCI is a static index over a one-year period. Conversely, intercounty human flow from SafeGraph is dynamic and can change dramatically, even within a week. We generated daily FCI (and FPC) for each county pair in the United States to utilize the full temporal resolution of the SDM dataset. We used weekly change (Δ) of FPC for the four temporally lagged weeks, and its average only in the fourth week as features -SG and -SGR models, with the same rationale as features in categories C and D to capture SIR compartments in connected counties (Table 5).

## Model implementation.
The ultimate target variable in all of our autoregressive models in each prediction horizon is the number of new cases of COVID-19 during that horizon. For training and tuning the models, however, we used a transformed target variable, namely the natural log-transformed values of new cases per 10k population plus one (to avoid zero values). For reporting the model predictions, we computed the number of new cases by applying an inverse transformation, i.e., an exponential transformation minus one (Formula 5–7). The rationale for using the log-transformed target variable, as opposed to directly predicting the weekly new cases, was to minimize skewness, and more importantly, minimize the sensitivity of the models to the population of counties. Our exploratory work did confirm that using this logged of incidence rates that produced better results.

$$y_{\text{predicted}(i,t)} = \ln(\Delta \text{ incidence rate}_{(i,t)} + 1) \quad (5)$$

$$\Delta \text{ incidence rate}_{\text{predicted}(i,t)} = e^{y_{\text{predicted}(i,t)}} - 1 \quad (6)$$

$$\Delta \text{ Case}_{\text{predicted}(i,t)} = (\Delta \text{ incidence rate}_{\text{predicted}(i,t)}) * \text{Population}_i / 10,000 \quad (8)$$

Δ Case$_{\text{predicted}(i,t)}$ in 7 denotes the number of new cases in county $i$ over the prediction horizon $t$.

Our training dataset includes up to 43 training samples per county (number of total samples $n = 3103 \times 43$), with each sample holding various features in one- to four-weekly temporal lags (Table 5). The weekly calculation of features is based on weeks starting on Sundays and ending on Saturdays, with predictions also made for horizons spanning Sunday–Saturday periods (both days are inclusive) as in common practice[25]. Our features, models, evaluations, and comparisons are limited to the counties in the coterminous United States. Table 5 summarizes the features that we used and the number of temporal lags (if any) used for each feature. All features were standardized for use in machine-learning algorithms.

Our general approach to training, validation, and testing of our models for different prediction horizons is similar, only, with target variables calculated separately for the specific prediction horizon. We first outline our approach for one-week ahead prediction horizons, which is used as the basis for algorithm selection. We then provide an overview of the implementation of the models for longer-term prediction horizons.

We trained and tuned the models using randomized search and 5-fold cross-validation, and tested the best-tuned model for predicting new cases on unseen data, during the n weeks following the forecast date where n is the prediction horizon: $n \in \{1, 2, 3, 4\}$. Cross-validation helps in preventing overfitting to a large degree[56].

For instance, for the forecast date of 2020/10/25, we used features that were generated using data collected until 2020/10/24 for training and tuning. The tuned model was then used for predicting new cases in each county during the 2020/10/25–2020/10/31 period. We used the reported cases by the JHU CSSE. The temporally lagged features for this forecast date were generated for t-1, t-2, t-3, and t-4 weekly lags, namely, the weeks ending on 2020/10/24, 2020/10/17, 2020/10/10, and 2020/10/03, respectively.

For the next forecast date, 2020/11/1, the training size increased by one week (per county), and the target week was also shifted by one week. Supplementary Table 1 summarizes the forecast dates, one-week and 4-week ahead prediction horizons, and training-data size. The data used in generating these features span a period from 2020/03/29 to 2021/01/23 to cover the temporal lags. This was the latest date for which 4-week ahead ground-truth data were available at the time of performing our analyses. Consequently, the target variable is collected through 2021/02/20 for the evaluation of four-week ahead predictions on the last forecast date. Our 14-week evaluation period covers both increasing and decreasing trends in the number of new cases in the United States, as well as the three highest peaks in the daily and weekly numbers of new cases in the country. More details on cross-validation, hyperparameters, and evaluation are presented in Supplementary Note 3.

We experimented with five different supervised machine-learning regression algorithms, namely Random Forest[57] (RF), Stochastic Gradient Boosting[58] (SGB), eXtreme Gradient Boosting[41,59] (XGB), Feed Forward Neural Network[60] (FFNN), and Long Short-Term Memory[61] (LSTM) network to build the autoregressive machine learning models with features described in Table 5. We evaluated the models using the dates listed in Supplementary Table 1. The results are presented in Table 1. The details of hyperparameter candidates and specific architectures are presented in Supplementary Note 3.

## Comparing Facebook-derived features with SafeGraph-derived features.
Since the XGB algorithm performed best (Table 1), we chose it as the selected machine learning algorithm, and trained the base, -FB, -SG, and -SGR models using the XGB algorithm to predict new cases of COVID-19 in short-term (one week) and long-term (two-four weeks) prediction horizons. To name a specific model in this article, we use a prefix that denotes the type of lag included in the model features (i.e., T for temporal or ST for spatiotemporal), followed by the name of the algorithm (XGB), followed by a suffix denoting the features included in the model, namely, -FB, -SG, and -SGR. Thus, TXGB (Temporal eXtreme Gradient Boosting) denotes the model that is built using the XGB algorithm and includes the base, temporally lagged features; and STXGB-FB (SpatioTemporal eXtreme Gradient Boosting) denotes the model that includes Facebook-derived features (and thus, spatiotemporal lags) and is built using XGB.

We evaluated the performance of TXGB, STXGB-FB, STXGB-SG, and STXGB-SGR by comparing the RMSE and MAE scores of the predictions against the observed numbers of new cases in the corresponding prediction horizon (the results are presented in Table 2 and Fig. 1).

Each STXGB model was tuned and trained on a regular desktop machine (with a 6-core Ryzen 5 3600X CPU and 64GB of RAM) in approximately 12–13 minutes for a single prediction horizon, and thus, in almost one hour for all of the four prediction horizons.

## Evaluation against the COVIDhub-Baseline and COVIDhub-Ensemble models.
In addition to the one-week short-term predictions, we performed long-term predictions of new COVID-19 cases in two-, three-, and four-week ahead prediction horizons. We only used the STXGB algorithm to develop long-term prediction models since it outperformed other algorithms in short-term predictions (see the Results section). We used the same set of features for long-term predictions, with modifications on the target variable to reflect different prediction

horizons. For instance, the target dates for two-, three-, and four-week ahead horizons of the Forecast date 2020/10/25, were 2020/11/07, 2020/11/14, and 2020/11/21, respectively.

The model for each horizon was trained and validated separately using the same training data and approach described in the previous section and was tested on two, three, and four weeks of unseen data, respectively, for each horizon. We evaluated the models' predictions by comparing them against the predictions generated by the COVIDhub-Baseline and -Ensemble models, as well as the ground-truth values of new cases derived from JHU CSSE COVID-19 reports.

**Generating prediction intervals using STXGB model**. To assess the uncertainty in the predictions generated using the STXGB models, we performed quantile regression on two quantiles, namely 2.5% (alpha = 0.025) and 97.5% (alpha = 0.975) using the quantile loss function of SGB, and calculated the 95% prediction interval as a result. These quantiles and the subsequent prediction interval are also reported for the COVIDhub models (and by the CDC[27]) and thus allow for comparing prediction intervals as well as point estimates between STXGB and those models (Fig. 3). We summed all the lower-quantile (2.5%) predicted values for all the counties to calculate the national lower-quantile predictions, and performed a similar process to calculate the national upper-quantile (97.5%) predictions.

**Reporting summary**. Further information on research design is available in the Nature Research Reporting Summary linked to this article.

## Data availability

All of the raw data used in this study are publicly available (at the time of writing this paper). We created socioeconomic features from the 5-year survey data—between 2014 and 2018—provided by the American Community Survey (ACS) and available at the IPUMS National Historical GIS portal (https://www.nhgis.org/). Daily maximum and minimum temperature surfaces of the United States published by the NOAA are available at https://ftp.cpc.ncep.noaa.gov/GIS/GRADS_GIS/GeoTIFF/TEMP/. We used the cumulative confirmed COVID-19 cases published by the Johns Hopkins University Center for Systems Science and Engineering (JHU CSSE) to generate COVID-related features. Facebook's Social Connectedness Index (SCI) database is available at https://dataforgood.fb.com/tools/social-connectedness-index/ and the movement-range dataset can be found at https://data.humdata.org/dataset/movement-range-maps. Finally, the instructions for accessing SafeGraph's Social Distancing Metrics dataset are available at https://docs.safegraph.com/docs/social-distancing-metrics.

The processed data used in this study have been deposited in the Zenodo database under accession code https://zenodo.org/record/5542643[62]. The processed data are available without any restriction. The raw data are also publicly available using links provided above.

## Code availability

All code necessary for the replication of our results are publicly available at https://github.com/geohai/COVID19-STGXB[63].

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

## Acknowledgements

This work was supported by the Population Council and the University of Colorado Population Center (CUPC) funded by Eunice Kennedy Shriver National Institute of Child Health & Human Development of the National Institutes of Health (P2CHD066613). The content is solely the responsibility of the authors, and does not reflect the views of the Population Council, or official views of the NIH, CUPC, or the University of Colorado. Publication of this article was partially funded by the University of Colorado Boulder Libraries Open Access Fund.

## Author contributions

MK conceptualized the project, designed the features, and contributed 30% of data processing and implementation, and contributed equally to writing. BV conducted the majority of data processing, implementation, literature review, and contributed equally to writing. HZ contributed to the study design and 10% of implementation and writing.

## Competing interests

The authors declare no competing interests.
