## [Peer Review File · Nature Communications]

Reviewers' Comments:

Reviewer #1:

Remarks to the Author:

This study combined the prediction power of Facebook's social connectedness with cell phone-derived human mobility in order to predict county-level new cases of COVID-19. The authors demonstrated a better proxy of social connectedness for measuring human interactions leads to new infections, and they developed a SpatioTemporal autoregressive eXtreme Gradient Boosting (STXGB) model to predict the county-level new cases of COVID-19 in the US. The paper is well-structured. However, there is a lack of methodological innovations and there are some concerns about the main concept, methodology, and interpretations as follows.

Main comments:

1. Although this paper stated that incorporating social media features (Facebook) would achieve better performance compared with cell phone-derived features (SafeGraph), it is still unclear whether the difference is caused by the data quality or the difference in the merit of these two datasets. In addition, the paper created an index while using SafeGraph data instead of directly introducing human mobility data. It seems to be a compromise with the incompleteness of Facebook data. This comparison is unfair.
2. The authors conducted experiments with five different machine learning-based spatiotemporal autoregressive algorithms to perform the county-level predictions, and then selected the best performing one. However, could they also discriminate which one of social connectedness data (social media) and human flow data (cell phone) contributes more to the prediction? The case may be varied in different counties using different models; it would be of great interest to provide corresponding interpretations about these differences across counties, e.g., the relevance with different social distance policies or different awareness of COVID severity among citizens in different counties.
3. Concerns arise immediately with the consideration that the elderly who are more vulnerable to COVID-19 infectious but do not frequently use social media apps.
4. Confounding variables exist such as seasonality and different control measures in different counties.
5. How did the authors deal with the fake connections in social media data (which are not uncommon)? Whether or not the high fake connection rate may lead to a less accurate prediction?
6. How do the authors ensure the model didn't provide overfitted predictions?
7. Could the authors provide some thoughts about model interpretability? i.e., the ranking of prediction variables in the final tree-based model, and that of interaction effects? In this case, the results could be more practically interpreted for better decision-making since a subset of variables would be enough to provide acceptable predictions in different counties.

Minor comments:

1. Limitations should be mentioned, e.g., less frequently usage rate of social media apps in some counties.
2. Table 3 provided results of prediction improvement using the best model over baselines, however, it would be better to demonstrate this significance with statistics tests, and confidence intervals shall be presented.

Reviewer #2:

Remarks to the Author:

The authors propose a regression model to predict county-level COVID-19 incidence. A range of features is designed based on combination of (i) previously reported incidence cases; (ii)

socioeconomic, demographic and environmental factors within county; and (iii) previously reported cases from coterminous counties weighted by social interaction indicators.

Major comments:

- I would suggest to change the second part of the title. It says: "Modeling Human Interactions using Social Media and Cell-Phone Data", but the manuscript does not explicitly model human interaction, only uses derived metrics from Facebook and SafeBoot as proxies.
- Is there any reason why this particular period (October 24- November 28 2020) was chosen for analysis? During this period, US national incident cases were increasing almost in a linear fashion.
- Line 378 states: " we also included (natural log-transformed values of) change in incidence rate, i.e., the subtraction of observed start-of-week incident rate from end-of-week incidence rate during the four weekly temporal lags". What would happen if number of new cases decreases and You have a logarithm of negative number?
- Statement on line 95 is too strong, i.e. "improving the long-term prediction of county-level new cases of COVID-19 in the coterminous U.S. in comparison to a baseline Ensemble model, using an end-to-end model. ". Maybe "comparing" would be more adequate.
- Please add how CDC ensemble forecast data were obtained, i.e. [1] or Github?
- Section "Long-term predictions and evaluation against the COVID-19 Forecast Hub Ensemble" would benefit from visual comparison of two methods. Could You make few figures similar to those produced by CDC, where recorded weekly cases are followed by week oone-four forecasts? I would like to see new cases aggregated over the whole US and some counties with high number of cases were STXGB-FB performed well, and Ensemble not.

Minor comments:

- Line 18: " Our experiments show that social connectedness..." These are not experiments, but analysis.
- Line 83: Give what acronyms RMSE (root mean squared error) and MAE (mean absolute error) stand for.
- Why Table 1 shows RMSE/MAE of a transformed target variable, while the rest of the analysis (Tables 2, 3, Figures 1, 2) show RMSE/MAE of target variable? It would be easier to read if all results would be given using the same metrics.

[1] <https://www.cdc.gov/coronavirus/2019-ncov/covid-data/forecasting-us-cases-previous.html>

Reviewer #3:

Remarks to the Author:

The authors applied machine-learning spatiotemporal algorithms to predict the transmission of SARS-COV-2 in the US. They present an interesting and novel computationally efficient way to predict new cases on the county level, but the incorporated dynamics have fundamental caveats and the results are not put in perspective. I do have considerable comments on the manuscript and methods.

The predictive value of the techniques is challenged over 5 weeks (October 24th until November 21st), but there is no information about the state of the epidemic in the US this period. Is there any sudden change in this period, or are the general trends in general up- or downwards? In the latter case, it is "relatively easy" to predict the future. The presented maps (Figure 3) do not provide enough information on the (predicted) trends in the number of cases. According the figures on the Covid-19 ForecastHub website, this period is marked by a slow increase in the number of cases. A reliable prediction model should be able to predict sudden increases, the peak of an epidemic wave and when declining trends will stabilize. The current manuscript does not include a thorough test of the algorithms for different epidemic situations.

Social contact behavior, which the authors clearly denote as the driving force of transmission dynamics, has changed dramatically during the COVID-19 pandemic on voluntary bases and because of restrictions. Changes in contact behavior altered the disease transmission and are linked with (large) outbreaks and declining incidences. As such, it seems counter intuitive to use a fixed "Social Connectedness Index" from Facebook's friendship data to predict adaptive social contact behavior. In addition, the hypothesis is not based on previous literature, according the introduction, and it might be that other confounders are at stake. What about age? Age has been proven as a driving force in our social contact behavior (Mossong 2008, Plos Med). There is not much information available on the population characteristics of Facebook friendship data and whether this is representative for the US.

The number of new cases is the most unstable predictor of the epidemic, since testing policies have been changed during the pandemic. Testing might increase if (local) authorities are suspicious about new outbreaks, and decrease if the system collapses or if the likelihood of being positive when experience symptoms is very high. Are the predictive models also able to capture trends in hospital admissions and mortality, which are relatively speaking more stable?

A comparison with top 10 models of the COVID-19 ForecastHub initiative would be more informative.

Reviewer #4:

Remarks to the Author:

In this paper, the authors carry out an impressive effort aimed at forecasting the spread of COVID-19 with machine learning models based on different proxies for human inter-connectedness. In particular, they compare the performance of models based on Facebook's social connectedness with cell phone-derived human mobility for predicting county level new cases of COVID-19. Then they develop a SpatioTemporal autoregressive eXtreme Gradient Boosting model to predict county-level new cases of COVID-19 comparing their predictions with a baseline Ensemble of 32-models currently used by the CDC for several temporal horizons. They claim an average 58% improvement in prediction RMSEs over two- to four week prediction horizons.

The effort to forecast covid-19 on a weekly basis and at a county level is something that has been carried out by many teams around the world, including the CDC Ensemble of 32 models mentioned as baseline for the predictions. The novelty of this work is in the comparison of different proxy data for human connectedness which result in similar performances (with respect to normalized RMSE and MAE) but with significant improvement with respect to the baseline. For long term predictions, the improvement with respect to the Ensemble CDC model is significant, especially for the 4-weeks temporal horizon.

The work falls in the vast research area aimed at providing effective tools for spatio-temporal forecasts of infectious diseases and contributes with a significant improvement with respect to state-of-the-art methodologies (e.g. the Ensemble approach used by CDC). On the other hand, it falls short in providing more explanatory information on why the social-media based connectedness and the mobile-phones one provide similar performances for predicting county level new cases of COVID-19 while the Facebook-based connectedness outperforms all the other models for the long-term predictions. This would go along the direction of explainable machine learning models. The authors devote a good effort in providing details about the proxy data (the Facebook's Social Connectedness Index and the cell-phone derived human mobility) but for, public health purposes, it would be interesting to have more insight on why a specific proxy dataset is more effective for forecasting purposes and possibly interventions design.

If the authors devoted some effort in providing a more accurate assessment on the possible determinants for the results they observe, the paper could be potentially highly influential for this field.

The data used in this work are public and methods are accurately described. This should ensure

the reproducibility of the work.

Statement of Revision

We would like to thank the reviewers and appreciate their thoughtful and constructive comments. In the following, we provide a detailed account of all the changes that we have made in the revised version of the paper. We have expanded our evaluation period from 5 weeks to 14 weeks (over 4 different prediction horizons, totaling 56 predictions and comparisons) to include both increasing and decreasing trends of the number of new COVID-19 cases. Also, in this revision, we have used the COVIDhub-baseline (Baseline) model in addition to the COVIDhub-ensemble (Ensemble) model as two different baselines for comparison, while ensuring that we are using exactly the same amount of data in training that would have been available for generating the Ensemble or Baseline prediction on each forecast date. We have also added prediction intervals and new visualizations to the paper, as suggested by the reviewers. We have structured this list in separate blocks corresponding to the comments made by the referees. The changes that have been made in the revised manuscript are in red followed by their page number in the revised manuscript.

Reviewer #1 (Remarks to the Author):

Comment 1. This study combined the prediction power of Facebook's social connectedness with cell phone-derived human mobility in order to predict county-level new cases of COVID-19. The authors demonstrated a better proxy of social connectedness for measuring human interactions leads to new infections, and they developed a SpatioTemporal autoregressive eXtreme Gradient Boosting (STXGB) model to predict the county-level new cases of COVID-19 in the US. The paper is well-structured. However, there is a lack of methodological innovations and there are some concerns about the main concept, methodology, and interpretations as follows.

Response . We have clarified the contributions of the paper in the Introduction section of the revised manuscript, on page 5. Additionally, we significantly revised the manuscript by expanding our evaluation period, comparing our models against COVIDhub-baseline model (in addition to the COVIDhub-ensemble model), and adding information on interpretability as described in the following responses.

To clarify, our methodological innovations include the quantitative capturing and incorporation of spatiotemporal lags in a replicable and efficient machine learning framework for Spatiotemporal autoregressive prediction of COVID-19 cases (which can be expanded to predict mortality rates or experimented for other diseases), in addition to comparison of Facebook-derived and SafeGraph-derived features. Our feature importance analysis added on page 31 of the revised article as well as in section E, page 15 of the supplementary information, also shows the high importance of spatiotemporal lags in predictive modeling of COVID-19 regardless of the specific feature set being used. We believe this is an important contribution, especially that Facebook Data is available for many countries, and that SafeGraph has shut down the Social Distancing dataset as of April 15, 2021. Additionally, our evaluations prove that both STXGB-FB and STXGB-SG models perform better than the COVIDhub-baseline model on average in two- to four-week prediction horizons, with

STXGB-SG (SafeGraph-derived) model outperforming the COVIDhub-ensemble model in three- and four-week prediction horizon as well.

Comment 2. Although this paper stated that incorporating social medial features (Facebook) would achieve better performance compared with cell phone-derived features (SafeGraph), it is still unclear whether the difference is caused by the data quality or the difference in the merit of these two datasets.

Response . Per this comment and to ensure a robust comparison between the two datasets, we expanded our evaluation period from 5 weeks to 14 weeks and performed a total of 56 predictions and comparisons over the four prediction horizons. The STXGB-FB model (which uses Facebook-derived features) outperformed the STXGB-SG model (which uses SafeGraph-derived features) in 36 out of 56 predictions (64.2% of predictions) and achieved lower average MAE in one- and two-week prediction horizons. These comparisons point to the overall advantage of Facebook-derived features (over SafeGraph derived features) in predicting COVID-19 cases in one- and two-week horizons, however, the more important message of our article is that Facebook-derived features can be used constructively for performant predictions. This is especially important for other countries, where Facebook data is available for. We have presented these comparisons in the Result section of the revised manuscript, pages 9-15, to clarify this point. That said, such a difference in the performance of the two feature sets might be a result of our model architecture and our conclusions are only valid within the framework that we have presented. We clarified this point in the Discussion section, page 29.

Furthermore, we added a section titled “Model Interpretability” to the revised version of the Supplementary Information document (section E, page 15) in which we analyzed the importance of all the features used in the STXGB-FB and STXGB-SG models. Please refer to our response to comment #4 for more details on this.

Comment 3. In addition, the paper created an index while using SafeGraph data instead of directly introducing human mobility data. It seems to be a compromise with the incompleteness of Facebook data. This comparison is unfair.

Response . We thank the Reviewer for this comment, however, we respectfully conjecture that they might have missed important information already in the paper. In the Methods section of the manuscript, page 46, we explain that in addition to Flow Proximity Index (derived from SafeGraph), we have also directly used SafeGraph-provided human mobility features in our -SG and -SGR models. These features include (mean and slope of) change in *completely_home_device_count* and *distance_traveled_from_home* in the -SG model, and *median_home_dwell_time* and *full_time_work_behavior_devices*, in addition to the aforementioned two features, in the -SGR model. Furthermore, we have also used Facebook-derived human movement features directly, including *Stay Put* and *Change in Movement*, in our -FB model. Therefore, we have indeed used “SafeGraph human mobility data” directly in our models.

The confusion might have stemmed from the (Social Proximity and Flow Proximity) indices that we have created for incorporating spatial lags, i.e., connection strength to other counties. Human mobility metrics are used directly in capturing movement within each county, in either Facebook- or SafeGraph-derived models

Comment 4. The authors conducted experiments with five different machine learning-based spatiotemporal autoregressive algorithms to perform the county-level predictions, and then selected the best performing one. However, could they also discriminate which one of social connectedness data (social media) and human flow data (cell phone) contributes more to the prediction? The case may be varied in different counties using different models; it would be of great interest to provide corresponding interpretations about these differences across counties, e.g., the relevance with different social distance policies or different awareness of COVID severity among citizens in different counties.

Response . Per this comment, we have added a section titled “Model Interpretability” to the revised version of the Supplementary Information document, on page 15. In this section, we have analyzed the importance of all the features used in the STXGB-FB and STXGB-SG models, across all forecasting dates and prediction horizons, by comparing the importance of the features. Based on this analysis, the importance of the Social Proximity to Cases (SPC) in the STXGB-FB model is higher than Facebook-derived movement features on average. One-week lagged change in SPC constantly has the second-highest importance in STXGB-FB model across all forecast dates/prediction horizons. On the contrary, Flow Proximity to Cases (FPC) does not achieve the same level of relative importance in the STXGB-SG model, and the importance of movement-related features is relatively higher in STXGB-SG compared to STXGB-FB.

In other words, the spatial lags captured using the Facebook-derived connectedness index consistently rank 2nd in the feature importance (right after the temporally lagged incidence rates) of STXGB-FB, while the SafeGraph-derived spatial lags are usually lower on the list of important features in STXGB-SG.

Section E of the Supplementary Information provides more detail on this analysis and Supplementary Fig. 3 and 4 demonstrate the feature importance of STXGB-FB and STXGB-SG models, respectively, for a snapshot on Nov. 8 forecasting date. It is important to note that because our models are implemented universally for all of the counties in the coterminous US, it is not possible to analyze feature importance at individual counties.

Comment 5. Concerns arise immediately with the consideration that the elderly who are more vulnerable to COVID-19 infectious but do not frequently use social media apps.

Response . As the Reviewer rightfully suggests, social media-derived data may not be as representative for the elderly. To directly account for the potential correlation between age and vulnerability to COVID-19, we added “*percentage of the population aged 65 or older*” as a new feature to all of our models and re-ran the models with this new feature. We believe adding this feature strengthened our models. Table 5 on page 39 of the revised manuscripts lists all the features that are used in our models.

However, it is important to note that our models are county-level predictions and do not make any age-specific or individual-level predictions. Both social media and cell-phone-derived data have age-related biases, as we explain in the Discussion section.

Comment 6. Confounding variables exist such as seasonality and different control measures in different counties.

Response . Per this comment, we have added a paragraph about seasonality and confounding variables in the revised manuscript, on page 38. In this paragraph, we have clarified that our models already capture seasonality to some extent by incorporating temporally lagged (a) temperature variables, (b) change in incidence rate (new cases), and (c) movement data (using both Facebook and SafeGraph data) as features. The variables observed in (b) and (c) are impacted by policy intervention or individual behavior change, manifested in data, therefore using these features we can capture the difference in control measures across space and time. Additionally, we use multiple weekly lags generated during the last 4-weeks leading to the forecast date to create a flexible predictive model that can capture changes in the season, policy, or trends, with non-linear XGB models.

Comment 7. How did the authors deal with the fake connections in social media data (which are not uncommon)? Whether or not the high fake connection rate may lead to a less accurate prediction?

Response . We have addressed this comment in the Discussion section of the revised manuscript, on page 34.

Facebook has more than 190 million active users in the US, the vast majority of which are not fake accounts, but as this comment rightfully suggests, fake connections do exist on social media data such as Facebook. By using county-level smoothed aggregates of Facebook connections to create our Facebook-derived features, we have tried to minimize the effect of fake connections. That said, machine learning algorithms can treat fake connections as noise (which is not learned by the model) automatically (and implicitly) when trained on the observed patterns of change in COVID-19 spread since such connections are not predictive of observed patterns.

Comment 8. How do the authors ensure the model didn't provide overfitted predictions?

Response . Per this comment, we added a brief clarification in the Methods section of the revised manuscript, on page 52. All of the tests that we have performed in this article have already been done on unseen data, over one- to four-week prediction horizons, to ensure overfitting is minimized. Plus, as previously mentioned in the "Model Implementation" subsection of the Methods section of the manuscript, "we trained and tuned the models using randomized search and 5-fold cross-validation". Cross-validation prevents overfitting to a large degree [1]. Only then, we evaluated the model on completely unseen data.

Comment 9. Could the authors provide some thoughts about model interpretability? i.e., the ranking of prediction variables in the final tree-based model, and that of interaction effects? In this case, the results could be more practically interpreted for better decision-making since a subset of variables would be enough to provide acceptable predictions in different counties.

Response . We have added a section titled "Model Interpretability" to the revised version of the Supplementary Information document, page 15, that includes the ranking of the importance of all the features used in the STXGB-FB and STXGB-SG model for an example forecasting date/prediction horizon. Based on the analysis provided, one-week lagged change in incidence rate ($\text{Ln}(\Delta \text{cumulative incidence rate} + 1)_{t-1}$) is the most important feature

in both models by a considerable margin. and the spatial lags derived from Facebook also consistently rank as the 2nd most important feature. We agree with the Reviewer that a subset of 10 features in each model may provide acceptable predictions, although we did not attempt to reduce the features.

Minor comments:

Comment 10. Limitations should be mentioned, e.g., less frequently usage rate of social media apps in some counties.

Response . We have added a subsection to the Discussion section in which we have addressed the abovementioned limitations as well as other general limitations of our model and study design. Pages 33 and 34 of the revised manuscript present the limitations.

Comment 11. Table 3 provided results of prediction improvement using the best model over baselines, however, it would be better to demonstrate this significance with statistics tests, and confidence intervals shall be presented.

Response . Per this suggestion, we re-implemented our models to generate prediction intervals and added a subsection to the Results section of the revised manuscript (pages 15-18) in which we compare the prediction intervals of the STXGB-FB and STXGB-SG models with the COVIDhub-baseline model at the 95% significance level. The Methods section of the revised manuscript describes the method we used to generate the prediction intervals. Furthermore, we have added a similar comparison, between STXGB-FB and STXGB-SG models with the COVIDhub-ensemble model in the Supplementary Information, pages 12-14.

Reviewer #2 (Remarks to the Author):

The authors propose a regression model to predict county-level COVID-19 incidence. A range of features is designed based on combination of (i) previously reported incidence cases; (ii) socioeconomic, demographic and environmental factors within county; and (iii) previously reported cases from coterminous counties weighted by social interaction indicators.

Major comments:

Comment 1. I would suggest to change the second part of the title. It says: "Modeling Human Interactions using Social Media and Cell-Phone Data", but the manuscript does not explicitly model human interaction, only uses derived metrics from Facebook and SafeBoot as proxies.

Response . We do understand their concern and have changed the title to "Predicting County-Level COVID-19 Cases using Spatiotemporal Machine Learning Through Social Media and Cell-Phone Data as Proxies For Human Interactions" in the revised manuscript. In our paper and definitions, the Social Proximity to Cases (SPC) and Flow Proximity to Cases (FPC) indices are proxies of human interaction, captured by social connectedness and flow connectedness respectively as we had mentioned in the original manuscript.

Comment 2. Is there any reason why this particular period (October 24- November 28 2020) was chosen for analysis? During this period, US national incident cases were increasing almost in a linear fashion.

Response . At the time of our original analyses, this period was the latest period for which we had access to 4-weeks ahead ground-truth data (confirmed number of COVID-19 cases were only available until Dec.19, 2020, at the time of our analysis). It is important to note that October 24 to November 21 was the period over which we generated predictions for one-, two-, three-, and four-week ahead horizons. Therefore, the target date for the four-week ahead predictions on the last forecast date was the week of Dec. 12 to Dec.19, 2020.

In the revised manuscript, we extended the evaluation period from 5 weeks to 14 weeks, covering the period between October 25, 2020, and January 24, 2021. The reason behind using this period as the evaluation period in our revised analysis is similar to what is described above: January 24, 2021 forecast date was the latest date for which we had access to four-week ahead confirmed number of COVID-19 cases. With the forecast dates covering the period between October 25, 2020, and January 24, 2021, the target date of the one-week ahead predictions covers the period from 2020-10-31 to 2021-01-30, and the target date of the four-week ahead predictions covers the period between **2020-11-21 and 2021-02-20**.

These dates are summarized in **Table 6** of the revised manuscript, **page 53**.

Comment 3. Line 378 states: " we also included (natural log-transformed values of) change in incidence rate, i.e., the subtraction of observed start-of-week incident rate from end-of-week incidence rate during the four weekly temporal lags". What would happen if number of new cases decreases and You have a logarithm of negative number?

Response . This was a mistake in our writing which we have fixed in the revised manuscript by modifying the definition of incidence rate on **page 37** of the revised manuscript.

We had already stated in the Methods section of the article (Base Features subsection) that: "we defined the COVID-19 incidence rate of a county as its number of cases per 10,000 population". We should have clarified that this is the cumulative incidence rates, as we also had stated in the supplementary Information, section A.3 "To measure county-level incidence rates at time t , we used the number cumulative confirmed COVID-19 cases at t , published by the Johns Hopkins University's Center for Systems Science and Engineering (JHU CSSE)." Therefore, when there are no new cases during a period, this value (change in cumulative incidence rates) will be zero. And since we use $\text{Ln}(\Delta \text{cumulative incidence rate}_{t+1})$ in the models, the value of this feature in this case (when there are no new cases) will be $\text{Ln}(0+1) = 0$.

Comment 4. Statement on line 95 is too strong, i.e. "improving the long-term prediction of county-level new cases of COVID-19 in the coterminous U.S. in comparison to a baseline Ensemble model, using an end-to-end model. ". Maybe "comparing" would be more adequate.

Response . We have addressed this point throughout the revised manuscript. Furthermore, we avoided using similar language in the revised manuscript whenever comparing the two models.

Comment 5. Please add how CDC ensemble forecast data were obtained, i.e. [1] or Github?

Response . The COVIDhub-ensemble (and baseline) forecast data that we have used in this paper are at the county level and have been downloaded from the COVID-19 Forecast Hub GitHub repository [2].

We have added references to both sources in the revised manuscript.

Comment 6. Section "Long-term predictions and evaluation against the COVID-19 Forecast Hub Ensemble" would benefit from visual comparison of two methods. Could You make few figures similar to those produced by CDC, where recorded weekly cases are followed by week one-four forecasts?

Response . Per this suggestion, we have generated new figures for both comparisons that show the observed number of new cases along with the predicted values generated by our models and the COVID-Hub models on each prediction horizon. Fig. 4 and Fig. 5 on pages 17 and 18 of the revised article show the observed and predicted values as well as prediction intervals generated by the STXGB-FB and STXGB-SG models in comparison with the COVIDhub-baseline respectively, for one- to four-week prediction horizons. Supplementary Fig. 2 and Supplementary Fig. 3 on pages 13 and 14 of the revised Supplementary Information document contain the same information for the Ensemble model instead of the Baseline.

Comment 7. I would like to see new cases aggregated over the whole US and some counties with high number of cases were STXGB-FB performed well, and Ensemble not.

Response . We have addressed this comment in the revised Supplementary Information with minor modifications: in addition to the national comparisons provided in the article (and mentioned in response to the previous comment), we performed model comparisons over 50 counties with the highest number of new cases during the one-week ahead of each forecast date.

We improved the manuscript by adding a section titled "Model comparison in counties with the highest number of new cases" on page 20 of the revised Supplementary Information.

Minor comments:

Comment 8. Line 18: " Our experiments show that social connectedness..." These are not experiments, but analysis.

Response . We have addressed this comment by replacing "experiments" with "analyses" in the revised manuscript.

Comment 9. Line 83: Give what acronyms RMSE (root mean squared error) and MAE (mean absolute error) stand for.

Response . We have added the expanded form of these two terms to the revised manuscript.

Comment 10. Why Table 1 shows RMSE/MAE of a transformed target variable, while the rest of the analysis (Tables 2, 3, Figures 1, 2) show RMSE/MAE of target variable? It would be easier to read if all results would be given using the same metrics.

Response . We appreciate this question and agree that it would have been easier to present training performance at the same scale. However, as mentioned in the article, we used “the natural log-transformed values of new cases per 10k population plus one” as the transformed target variable to train our models and the values presented in Table 1 are calculated as part of, and during, the training procedure and by aggregating error over all counties. The decision to select the best model is based on this transformed target variable (whose errors are minimized during the training process), therefore, we decided that listing the transformed variables errors reflects that our model selection criteria the best

Reviewer #3 (Remarks to the Author):

The authors applied machine-learning spatiotemporal algorithms to predict the transmission of SARS-COV-2 in the US. They present an interesting and novel computationally efficient way to predict new cases on the county level, but the incorporated dynamics have fundamental caveats and the results are not put in perspective. I do have considerable comments on the manuscript and methods.

Comment 1. The predictive value of the techniques is challenged over 5 weeks (October 24th until November 21st), but there is no information about the state of the epidemic in the US this period. Is there any sudden change in this period, or are the general trends in general up- or downwards? In the latter case, it is “relatively easy” to predict the future. The presented maps (Figure 3) do not provide enough information on the (predicted) trends in the number of cases. According the figures on the Covid-19 ForecastHub website, this period is marked by a slow increase in the number of cases. A reliable prediction model should be able to predict sudden increases, the peak of an epidemic wave and when declining trends will stabilize. The current manuscript does not include a thorough test of the algorithms for different epidemic situations.

Response . October 24 to November 21 was the original period for which we generated predictions over one-, two-, three-, and four-week ahead horizons. Therefore, the target date for the four-week ahead predictions on the last forecast date (before revisions) was the week of Dec. 12 to Dec.19, 2020, and this period did also include decreasing weekly trends of new cases in the week of November 21 to November 28 (Figure 1). Per this comment, in the revised manuscript, we extended the evaluation period from 5 weeks to 14 weeks, covering the period between October 25, 2020, and January 24, 2021, and clarified the status of the pandemic in the revised manuscript (“Long-term predictions and evaluation against the COVIDhub-Baseline” subsection and Figure 3).

The reason behind using this extended evaluation period in our revised analysis is to include both the increasing and decreasing as well as the peak of the pandemic. The January 24,

2021 forecast date was the latest date for which we had access to the four-week ahead confirmed number of COVID-19 cases when we performed our analyses for the revised manuscript. With the extended forecast dates covering the period between October 25, 2020, and January 24, 2021, the target date of the one-week ahead predictions covers the period from 2020-10-31 to 2021-01-30, and the target date of the four-week ahead predictions covers the period between 2020-11-21 and 2021-02-20. This extended period contains periods of sudden increases, the national peak of the epidemic, and declining trends (Figure 1).

Furthermore, in the revised manuscript, we compare our models with the COVIDhub-baseline model and we show that both our STXGB-FB and STXGB-SG models perform better than the Baseline on average in two-, three-, and four-week prediction horizons during our 14-week evaluation period (Table 3 on page 14 of the revised manuscript). The baseline model performs better than both our STXGB-FB and STXGB-SG models on most of the forecast dates/prediction horizons during the period between 2020-12-06 and 2021-01-03 (5 forecast dates) which marks the second-highest peak in the number of daily new cases as well as an increasing and decreasing trend (Table 3, Fig. 2 and Fig. 3 in the revised manuscript). On the other 9 forecast dates before and after this period, in 35 out of 36 forecast date/prediction horizons, at least one of our models outperforms the COVIDhub-Baseline. This shows that our models outperform the Baseline during steady increase/decrease trends, but do not perform as well during periods of rapid fluctuations. It is important to note that these periods of fluctuation are marked by underreporting on holidays (e.g. Christmas, Thanksgiving, or New Year), followed by overreporting post-holidays, and therefore, the model comparisons during these weeks might not be reflective of their true performance.

We believe evaluating model predictions against 7-day smoothed values would provide a better reflection of model performance and might benefit our models. However, in this study, we chose to evaluate against the weekly aggregation of raw numbers (even with inconsistencies) to ensure that we are not characterizing our results with any additional advantage. We have mentioned these in the Discussion section of the revised article, on page 22.

Figure 1. Daily numbers of new COVID-19 cases (7-day averaged). from New York Times [4].

Comment 2. Social contact behavior, which the authors clearly denote as the driving force of transmission dynamics, has changed dramatically during the COVID-19 pandemic on voluntary bases and because of restrictions. Changes in contact behavior altered the disease transmission and are linked with (large) outbreaks and declining incidences. As such, it seems counter intuitive to use a fixed “Social Connectedness Index” from Facebook’s friendship data to predict adaptive social contact behavior.

Response . We have clarified how the dynamic aspects of social distancing are reflected in our models, in the revised manuscript (Methods section).

As we had mentioned in the Methods section of the article, in addition to Social Connectedness Index (SCI), we have used daily movement data published by Facebook (*stay put* and *change in movement* features, Table 4 in the original manuscript) that are dynamic and capture changes in daily movements, either as a result of policies or personal choice. Furthermore, SCI is not used directly in our models as a feature, rather we have used SCI to calculate spatial lags of infection, which we call social proximity to cases (SPC). SPC is a measure of the level of exposure to COVID-19 cases in connected counties weighted by the inter-county social connectedness strength, therefore SPC is not a static measure either (as the number of cases changes weekly). Finally, SCI is measured and published twice a year by Facebook, and the reason it is a “static” index is that the relative number of friendships does not change significantly when aggregated nationally or at the county level.

Comment 3. In addition, the hypothesis is not based on previous literature, according the introduction, and it might be that other confounders are at stake. What about age? Age has been proven as a driving force in our social contact behavior (Mossong 2008, Plos Med). There is not much information available on the population characteristics of Facebook friendship data and whether this is representative for the US.

Response . Per this comment, we added a discussion on the representativeness of Facebook and cell phone data, as well as our work’s potential limitations to the revised manuscript, **in the Results section**.

Additionally, as part of our revisions and model rebuilding, we added “percentage of the population aged 65 or older” as a new feature to the demographic and socioeconomic features (category A features that are included in all of our models), citing Mossong 2008 as the rationale, and trained the models with this additional feature (**Table 5 on page 39 of the revised manuscript**). In the Introduction section of the article, we had cited previous works that have used Facebook- or SafeGraph-derived variables to study the spread of COVID-19. We had also cited related works that have used a similar set of socioeconomic, demographic, and climatic variables to predict COVID-19 (in the Supplementary Information).

Per this comment, we also added a discussion on the potential limitations related to the representativeness of Facebook data (social media data in general) to the revised manuscript, **on page 34**.

Comment 4. The number of new cases is the most unstable predictor of the epidemic, since testing policies have been changed during the pandemic. Testing might increase if (local) authorities are suspicious about new outbreaks, and decrease if the system collapses or if the likelihood of being positive when experience symptoms is very high. Are the predictive models also able to capture trends in hospital admissions and mortality, which are relatively speaking more stable?

Response . We thank the Reviewer for this comment, and agree with them regarding the potential inconsistencies in testing and reporting COVID-19. We had already mentioned and discussed the potential inconsistencies in testing and reporting in the Results and Discussion sections of the original manuscript and per this comment, we have further elaborated upon such shortcomings in the revised manuscript, on page 22.

However, such inconsistencies are not limited to the number of cases and do exist in the reported numbers of hospitalization and mortalities related to COVID-19. For instance, there is evidence that mortality in the US is also undercounted, spatially heterogeneous, and not measured with the same policy everywhere. Weinberget et. al [5] state in their conclusions that: “Excess deaths provide an estimate of the full COVID-19 burden and indicate that official tallies likely undercount deaths due to the virus. The mortality burden and the completeness of the tallies vary markedly between states.” A newer preprint in 2021 states that: “Higher per capita testing was associated with more complete reporting of COVID-19 deaths, which is a fundamental requirement for analyzing the pandemic.” [6].

In theory, our models would be capable of predicting hospitalization or mortality rates if they are trained on these variables as the target variable. However, it is out of the scope of our paper to perform a different prediction task. We have suggested this as a potential future work in the revised manuscript on page 32 to provide directions for the research community.

Comment 5. A comparison with top 10 models of the COVID-19 ForecastHub initiative would be more informative.

Response . We appreciate this suggestion. COVID-19 Forecast Hub has recently published a preprint in which they have compared the models that perform forecasts of incident mortality counts in the US [7]. Cramer et al. state in this paper that “comparative evaluation of the considered models is hampered by the fact that not all of them provide forecasts for the same set of locations and time points”. Furthermore, they mention that “all models showed large variability in skill relative to other models”. A fact that in turn makes model accuracy rankings “highly variable”.

To the best of our knowledge, however, COVID-19 Forecast Hub has not officially reported any ranking of the models that perform forecasts of reported weekly incident COVID-19 cases. As a result, identifying the “top 10 models of the COVID-19 Forecast Hub initiative” that perform forecasts of reported weekly incident COVID-19 cases is a challenging task that could be the topic of separate research, and thus is out of the scope of our paper. Furthermore, the “top models” would be different from week to week, making this task more subjective. More importantly, the Ensemble model is guaranteed to always perform better

than any individual model, and thus, we compare our model against the Ensemble model as well.

Reviewer #4 (Remarks to the Author):

In this paper, the authors carry out an impressive effort aimed at forecasting the spread of COVID-19 with machine learning models based on different proxies for human inter-connectedness. In particular, they compare the performance of models based on Facebook's social connectedness with cell phone-derived human mobility for predicting county level new cases of COVID-19. Then they develop a SpatioTemporal autoregressive eXtreme Gradient Boosting model to predict county-level new cases of COVID-19 comparing their predictions with a baseline Ensemble of 32-models currently used by the CDC for several temporal horizons. They claim an average 58% improvement in prediction RMSEs over two- to four week prediction horizons.

The effort to forecast covid-19 on a weekly basis and at a county level is something that has been carried out by many teams around the world, including the CDC Ensemble of 32 models mentioned as baseline for the predictions. The novelty of this work is in the comparison of different proxy data for human connectedness which result in similar performances (with respect to normalized RMSE and MAE) but with significant improvement with respect to the baseline. For long term predictions, the improvement with respect to the Ensemble CDC model is significant, especially for the 4-weeks temporal horizon.

Comment 1. The work falls in the vast research area aimed at providing effective tools for spatio-temporal forecasts of infectious diseases and contributes with a significant improvement with respect to state-of-the-art methodologies (e.g. the Ensemble approach used by CDC). On the other hand, it falls short in providing more explanatory information on why the social-media based connectedness and the mobile-phones one provide similar performances for predicting county level new cases of COVID-19 while the Facebook-based connectedness outperforms all the other models for the long-term predictions.

Response . We have added a section titled "Model Interpretability" to the revised Supplementary Information document, section E page 15. In this section, we have analyzed the importance of all the features used in the STXGB-FB and STXGB-SG models, across all forecasting dates and prediction horizons, by comparing the importance of the features. We have also added a paragraph to the Discussion section of the revised manuscript, on pages 30 and 31, in which we discuss the potential reasons behind the difference in the performance of the Facebook-derived features versus SafeGraph-derived features.

Comment 2. This would go along the direction of explainable machine learning models. The authors devote a good effort in providing details about the proxy data (the Facebook's Social Connectedness Index and the cell-phone derived human mobility) but for, public health purposes, it would be interesting to have more insight on why a specific proxy dataset is more effective for forecasting purposes and possibly interventions design. If the authors devoted some effort in providing a more accurate assessment on the possible determinants for the results they observe, the paper could be potentially highly influential for this field.

Response . Per this comment, and as mentioned in response to the previous comment, we added an analysis of the importance of all the features used in the STXGB-FB and STXGB-SG models, across all forecasting dates and prediction horizons, to provide interpretability for the wider scientific community (and potentially for the decision-makers). **Section E** of the Supplementary Information provides more detail on this analysis and **Supplementary Fig. 3 and 4** demonstrate feature importance of STXGB-FB and STXGB-SG models respectively, for a snapshot on Nov. 8 forecasting date.

In summary, based on the analysis of feature importance, the importance of the Social Proximity to Cases (SPC) in the STXGB-FB model is higher than Facebook-derived movement features on average. One-week lagged change in SPC constantly has the second-highest importance in STXGB-FB model across all forecast dates/prediction horizons, pointing to the strong predictive power of spatial lags captured using the Facebook-driven social proximity to cases. On the contrary, Flow Proximity to Cases (FPC) does not achieve the same level of relative importance in the STXGB-SG model, and the importance of movement-related features is relatively higher in STXGB-SG compared to STXGB-FB.

References

- [1] Hawkins, D. M. (2004). The problem of overfitting. *Journal of chemical information and computer sciences*, 44(1), 1-12.
- [2] COVID-19 Forecast Hub GitHub repository. DOI: 10.5281/zenodo.4305938. Accessed online on April 23, 2021, at <https://github.com/reichlab/covid19-forecast-hub/tree/master/data-processed>
- [3] Coronavirus in the U.S.: Latest Map and Case Count. *New York Times*. Accessed online on April 23, 2021, at <https://www.nytimes.com/interactive/2021/us/covid-cases.html>
- [4] Weinberger, D. M., Chen, J., Cohen, T., Crawford, F. W., Mostashari, F., Olson, D., ... & Viboud, C. (2020). Estimation of excess deaths associated with the COVID-19 pandemic in the United States, March to May 2020. *JAMA Internal Medicine*, 180(10), 1336-1344.
- [5] Perniciaro, S. R., & Weinberger, D. M. (2021). 50 policies, 1 pandemic, 500,000 deaths: Associations between state-level COVID-19 testing recommendations, tests per capita, undercounted deaths, vaccination policies, and doses per capita in the United States. *medRxiv*, 2020-09.
- [6] Cramer, E. Y., et al. (2021). Evaluation of individual and ensemble probabilistic forecasts of COVID-19 mortality in the US. *medRxiv*. (preprint)

Reviewers' Comments:

Reviewer #1:

Remarks to the Author:

The authors have significantly improved the paper with additional analysis of the model, comparison with baselines, and discussions of contributions. I would like to thank the authors for their hard work. The current paper is technically sound and clearly written.

However, given the huge amount of COVID-19 papers (many papers had similar ideas and methods and got published months ago - I actually reviewed quite a number of similar papers), I do not feel that the results are really noteworthy. In addition, the methods are not novel - there are tons of better models in CS conferences these two years. This paper is arguably the better-written one among similar papers that I have reviewed, but whether it is worth publishing in a Nature-branded journal is in doubt.

Reviewer #2:

Remarks to the Author:

The authors have satisfactorily responded to my comments and made the necessary changes to the manuscript.

My minor enquiry is about Fig. 4 and Fig. 5 showing the observed and predicted values. Legend says 'The solid black line represents the total number of observed cases at each forecast date.', but y-axis says 'Daily new cases (x weeks ahead)'. Could You plot predicted/observed number of new cases at forecast date rather than the sum of new cases during predicted horizon up to forecast date?

This could require revising the conclusion on lines 230-234.

Reviewer #3:

Remarks to the Author:

The authors did a great job in revising their manuscript according the comments and feedback from the reviewers. The extension of the evaluation period from 5 weeks to 14 weeks to include both increasing and decreasing trends in the number of new COVID-19 cases really improves the value of the work. Also, the comparison with the COVIDhub-baseline and COVIDhub-ensemble model expands the credibility of the work. Their point-by-point reply is very detailed, constructive and respectful. I recommend this paper for publication.

Reviewer #4:

Remarks to the Author:

The authors have significantly improved the quality and depth of the paper and have properly and extensively addressed my comments. I recommend for the paper to be published.

Statement of Revision (second round)

We would like to extend our sincere gratitude and appreciation to all of the reviewers for their feedback as well as their constructive remarks on this and the previous round of revisions. In the following, we provide a detailed account of all the changes that we have made in the revised version of the paper.

Reviewer #1 (Remarks to the Author):

Comment 1. The authors have significantly improved the paper with additional analysis of the model, comparison with baselines, and discussions of contributions. I would like to thank the authors for their hard work. The current paper is technically sound and clearly written. However, given the huge amount of COVID-19 papers (many papers had similar ideas and methods and got published months ago - I actually reviewed quite a number of similar papers), I do not feel that the results are really noteworthy. In addition, the methods are not novel - there are tons of better models in CS conferences these two years. This paper is arguably the better-written one among similar papers that I have reviewed, but whether it is worth publishing in a Nature-branded journal is in doubt.

Response . We would like to thank the reviewer for their comment and do not have any further response.

Reviewer #2 (Remarks to the Author):

Comment 1. The authors have satisfactorily responded to my comments and made the necessary changes to the manuscript. My minor enquiry is about Fig. 4 and Fig. 5 showing the observed and predicted values. Legend says 'The solid black line represents the total number of observed cases at each forecast date.', but y-axis says 'Daily new cases (x weeks ahead)'. Could You plot predicted/observed number of new cases at forecast date rather than the sum of new cases during predicted horizon up to forecast date? This could require revising the conclusion on lines 230-234.

Response . We would like to thank the reviewer for their comment and pointing us to this misleading label. The y-axis label was misleading since those figures represented the number of weekly new cases in n week(s) ahead when $n = 1,2,3,4$. Therefore, in the most recent revision we have changed that label.

On the other hand, given that forecast dates are all Sundays, plotting just new cases on a forecast date (rather than the sum of new cases during the week starting from the forecast date) would be misleading and exclude cases in other 6 days of the week. Therefore, we fixed the y-axis label and plotted the weekly news cases.

Reviewer #3 (Remarks to the Author):

Comment 1. The authors did a great job in revising their manuscript according the comments and feedback from the reviewers. The extension of the evaluation period from 5 weeks to 14 weeks to include both increasing and decreasing trends in the number of new COVID-19

cases really improves the value of the work. Also, the comparison with the COVIDhub-baseline and COVIDhub-ensemble model expands the credibility of the work. Their point-by-point reply is very detailed, constructive and respectful. I recommend this paper for publication.

Response . We would like to thank the reviewer for their comment and do not have any further response.

Reviewer #4 (Remarks to the Author):

Comment 1. The authors have significantly improved the quality and depth of the paper and have properly and extensively addressed my comments. I recommend for the paper to be published.

Response . We would like to thank the reviewer for their comment and do not have any further response.